

# Monogenean anchor morphometry: systematic value, phylogenetic signal, and evolution

Tsung Fei Khang[1], Oi Yoon Michelle Soo[2], Wooi Boon Tan[3] and Lee Hong Susan Lim[2]

[1] Institute of Mathematical Sciences, University of Malaya, Kuala Lumpur, Malaysia
[2] Institute of Biological Sciences, University of Malaya, Kuala Lumpur, Malaysia
[3] Centre for Tropical Biodiversity Research, University of Malaya, Kuala Lumpur, Malaysia

Corresponding author
Tsung Fei Khang,
tfkhang@um.edu.my

## ABSTRACT

**Background.** Anchors are one of the important attachment appendages for monogenean parasites. Common descent and evolutionary processes have left their mark on anchor morphometry, in the form of patterns of shape and size variation useful for systematic and evolutionary studies. When combined with morphological and molecular data, analysis of anchor morphometry can potentially answer a wide range of biological questions.

**Materials and Methods.** We used data from anchor morphometry, body size and morphology of 13 *Ligophorus* (Monogenea: Ancyrocephalidae) species infecting two marine mugilid (Teleostei: Mugilidae) fish hosts: *Moolgarda buchanani* (Bleeker) and *Liza subviridis* (Valenciennes) from Malaysia. Anchor shape and size data ($n = 530$) were generated using methods of geometric morphometrics. We used 28S rRNA, 18S rRNA, and ITS1 sequence data to infer a maximum likelihood phylogeny. We discriminated species using principal component and cluster analysis of shape data. Adams's $K_{mult}$ was used to detect phylogenetic signal in anchor shape. Phylogeny-correlated size and shape changes were investigated using continuous character mapping and directional statistics, respectively. We assessed morphological constraints in anchor morphometry using phylogenetic regression of anchor shape against body size and anchor size. Anchor morphological integration was studied using partial least squares method. The association between copulatory organ morphology and anchor shape and size in phylomorphospace was used to test the Rohde-Hobbs hypothesis. We created monogeneaGM, a new R package that integrates analyses of monogenean anchor geometric morphometric data with morphological and phylogenetic data.

**Results.** We discriminated 12 of the 13 *Ligophorus* species using anchor shape data. Significant phylogenetic signal was detected in anchor shape. Thus, we discovered new morphological characters based on anchor shaft shape, the length between the inner root point and the outer root point, and the length between the inner root point and the dent point. The species on *M. buchanani* evolved larger, more robust anchors; those on *L. subviridis* evolved smaller, more delicate anchors. Anchor shape and size were significantly correlated, suggesting constraints in anchor evolution. Tight integration between the root and the point compartments within anchors confirms the anchor as a single, fully integrated module. The correlation between male copulatory organ

morphology and size with anchor shape was consistent with predictions from the Rohde-Hobbs hypothesis.

**Conclusions.** Monogenean anchors are tightly integrated structures, and their shape variation correlates strongly with phylogeny, thus underscoring their value for systematic and evolutionary biology studies. Our `MonogeneaGM` R package provides tools for researchers to mine biological insights from geometric morphometric data of speciose monogenean genera.

## INTRODUCTION

The Monogenea is a class of flatworms (Platyhelminthes) that are primarily ectoparasites of fish (*Whittington, 2005*; *Hayward, 2005*). An adult monogenean parasite has well-developed attachment appendages located at its anterior (prohaptor) and posterior (opisthaptor) regions that help it to resist physical dislodgement from the host. The posterior attachment organs consist of sclerotized hard parts such as hooks, anchors and clamps. Ecologically, monogenean parasites are characterized by their strong host specificity (*Whittington et al. 2000*).

The Monogenea has several desirable features that make it invaluable as a model system for studying evolutionary processes that resulted in its past diversification and present diversity (*Poulin, 2002*). Primarily, many of its genera are speciose, morphologically diverse, show well-resolved phylogenies at the familial level (*Boeger & Kritsky, 1997*; *Boeger & Kritsky, 2001*; *Mollaret, Jamieson & Justine, 2000*), and samples can be easily obtained in large numbers. It has been used as a model to shed light on ecological forces that shape species community and structure (*Rohde, 1979*; *Mouillot et al., 2005*; *Raeymaekers et al., 2008*), to investigate processes leading to speciation and its maintenance (*Rohde & Hobbs, 1986*; *Rohde, 1994*; *De Meeus, Michalakis & Renaud, 1998*; *Šimková et al., 2002*; *Hahn et al., 2015*; *Vanhove & Huyse, 2015*), to elucidate host-parasite evolutionary ecology (*Desdevises et al., 2002*; *Huyse, Audenaert & Volckaert, 2003*; *Huyse & Volckaert, 2005*; *Šimková et al., 2006*; *Šimková & Morand, 2008*; *Mendlová & Šimková, 2014*; *Grégoir et al., 2015*), and to explore the extent of correlation between phenotype variation in attachment organs and factors such as phylogeny, host specificity and geographical location (*Vignon, Pariselle & Vanhove, 2011*).

Morphometric variation in anatomical structures of interest can be studied using two approaches. Traditional morphometrics (*Reyment, Blackith & Campbell, 1984*; *Marcus, 1990*) is characterized by the use of lengths of defined positions on anatomical structures of interest (or their ratios) as input data for multivariate statistical analyses. While such variables may measure size adequately, they are generally not effective for capturing shape information present in the geometry of a set of defined points of an

object (*Rohlf & Marcus, 1993*). A large proportion of biological variation due to shape differences is therefore missed when an analysis uses only information from variation in length variables.

With the development of geometric morphometrics over the past three decades, researchers now have, at their disposal, a powerful method for extracting, visualizing and combining shape data with other data types such as molecular phylogenies to attain an integrative evolutionary analysis (*Rohlf & Marcus, 1993*; *Adams, Rohlf & Slice, 2004*; *Adams, Rohlf & Slice, 2013*). Digitization of the anatomical structure of interest provides the key to the acquisition and use of a new type of data—landmark coordinates, from which shape information can be effectively extracted, and then analyzed, using new tools such as Procrustes superimposition, thin plate splines, relative warp analysis and elliptic Fourier analysis. Geometric morphometrics is now commonly used in systematics and evolutionary biology research where analysis of shape can be expected to provide new insights to complement traditional morphometric, phylogenetic or biogeographic analyses. A cursory search in major biological journal databases for recent publications having "geometric morphometrics" in their keywords revealed that geometric morphometrics is widely used to study various biological aspects, in diverse phyla, such as fish taxonomy (*Sidlauskas, Mol & Vari, 2011*), plant taxonomy (*Conesa, Mus & Rosselló, 2012*), gastropod shell shape variation (*Smith & Hendricks, 2013*; *Cruz, Pante & Rohlf, 2012*), morphological adaptation in birds (*Sievwright & MacLeod, 2012*), fly wing evolution (*Pepinelli, Spironello & Currie, 2013*), turtle neck shape evolution (*Werneburg et al., 2015*), beetle speciation (*Pizzo, Zagaria & Palestrini, 2013*) and species boundary problems in butterflies (*Barão et al., 2014*). Because of the inherently digital nature of geometric morphometric data, its increasing prominence in morphological studies accentuates the role of informatics in modern taxonomy (*Wheeler, 2007*).

In morphological analyses of monogeneans, taxonomists often prioritize prominent sclerotized parts such as the copulatory organ, because qualitative variation in the latter is frequently sharp and easy to describe. Nonetheless, morphometric variation in all sclerotized parts of monogeneans has been studied for a long time from the perspective of systematics (e.g., *Shinn, Gibson & Sommerville, 2001*) and evolutionary ecology (e.g., *Poisot & Desdevises, 2010*; *Mendlová & Šimková, 2014*). Hard parts such as the anchors are ideal for geometric morphometric analysis because they are not easily deformed by compression when mounted onto slides (*Lim & Gibson, 2009*). Anchor shape and size are taxonomically informative. Typically, size information is captured quantitatively in the form of distances between two defined points on the anchors, and shape information is captured qualitatively in the form of character states. The analysis of monogenean morphometric data has been, and continues to be, dominated by the application of traditional morphometrics (e.g., *Mariniello et al., 2004*; *Shinn et al., 2004*; *Tan, Khang & Lim, 2010*; *Hahn et al., 2011*; *Soo & Lim, 2012*). To date, there are only few examples (*Vignon & Sasal, 2010*; *Vignon, 2011*; *Vignon, Pariselle & Vanhove, 2011*; *Rodríguez-González et al., 2015a*) of applying geometric morphometrics to analyze monogenean anchor shape variation to overcome the limitations in data resolution inherent in standard morphometric and qualitative analyses. The paucity of geometric morphometric studies, however, belies the importance of this

approach in uncovering intraspecific shape variation in anchors that can be invaluable for species delimitation, particularly in resolving synonymies (e.g., *Pérez Ben, Gómez & Báez, 2014*), as well as for testing hypotheses of morphological integration (*Olson & Miller, 1958*) and evaluating levels of phenotypic plasticity (*Pfennig et al., 2010*).

As anchors serve a functional purpose, a priori, it is unclear whether phenotypic similarity of anchors among species is an outcome of adaptive processes related to the ecology or the morphology of their fish host, or simply a reflection of their phylogenetic constraint (*Morand et al., 2002*). If the presence of phylogenetic signal in anchors can be statistically established, evolutionary analysis of shape and size change can then be used to elucidate trends in particular clades. The results are expected to be useful for guiding the selection of appropriate anchor morphometric variables for conversion into morphological characters that have lower levels of homoplasy, thus overcoming the problem of unnecessary homoplasy of a morphological character arising from poor quality and insufficient number of character states (*Perkins et al., 2009*).

In this paper, we developed an integrative analysis that uses data from anchor morphometry and morphology, as well as DNA sequences, that allows the investigation of broad aspects in the systematic biology of monogeneans, such as species discrimination, evolutionary ecology, phylogenetic signal, and morphological integration. For illustration, we used data obtained from 13 recently described species belonging to the *Ligophorus* (Monogenea: Ancyrocephalidae) genus, a particularly speciose genus with 63 species known to date (Table S1). Thus, our study covered all the *Ligophorus* species (approximately 20% of global *Ligophorus* diversity) currently found on two mullet host species in Southeast Asia. Mullets are eaten by people in this region.

## MATERIALS AND METHODS

### Samples

We obtained 530 specimens (metadata in Table S2; sample sizes in Table S3) belonging to 13 *Ligophorus* species (*Soo & Lim, 2012*; *Soo & Lim, 2015*; *Soo, Tan & Lim, 2015*) from two species of adult mullet host: *Liza subviridis* Valenciennes, 1836 ($n = 52$) and *Moolgarda buchanani* Bleeker, 1853 ($n = 29$) from several locations in tropical Western Peninsular Malaysia (Fig. S1). The specimens have been deposited in the Museum of Zoology at University of Malaya (508 specimens), the Natural History Museum, London (14 specimens), and the Lee Kong Chian Natural History Museum, Singapore (6 specimens).

From 2009 to 2014, *Liza subviridis* was collected off Carey Island in Selangor (2°52′N, 101°22′E), and *M. buchanani* off Langkawi Island in Kedah (6°21′N, 99°48′E) and the sea off Johor (1°20′N, 103°32′E). The fish were obtained from local fishermen in the vicinity of the sampling locations, dead at the time of purchase. Seven of the 13 species: *L. bantingensis Soo & Lim, 2012*, *L. careyensis Soo & Lim, 2012*, *L. chelatus Soo & Lim, 2012*, *L. funnelus Soo & Lim, 2012*, *L. navjotsodhii Soo & Lim, 2012*, *L. parvicopulatrix Soo & Lim, 2012* and *L. belanaki Soo & Lim, 2015* were found on *Liza subviridis*; the remainder: *L. fenestrum Soo & Lim, 2012*, *L. kedahensis Soo & Lim, 2012*, *L. kederai Soo & Lim, 2015*, *L. grandis Soo, Tan & Lim, 2015*, *L. johorensis Soo, Tan & Lim, 2015*, and *L. liewi*

*Soo, Tan & Lim, 2015*, were found on *M. buchanani*. For all 81 fish examined, the *Ligophorus* species found in *M. buchanani* were never observed in *Liza subviridis*, and vice versa.

When preparing slides, we used a basic mounting protocol (*Lim, 1991*) where the monogeneans were put onto a clean slide with a drop of water, and then covered with a coverslip. Specimens were initially mounted in modified ammonium picrate glycerine, and subsequently converted into unstained, permanent mounts in Canada balsam. The opisthaptorial sclerotized hard parts of *Ligophorus* consist of a pair (left and right) of dorsal and ventral anchors, bars, and marginal hooks. Digital images of these hard parts were taken from labelled mounted slides using a light microscope with Leica digital camera (DFC 320) connected to the QWin plus image analysis software (Leica Microsystems, Germany) under 40x magnification, saved as jpeg files and organized into folders. Three species (*L. fenestrum*, *L. liewi* and *L. kederai*) showed a probable fixed character state—that of the presence of fenestrated structures on anchors of all examined specimens, since no variation in this character state was observed in all examined specimens of these three species.

## Data acquisition

Where possible, we measured a specimen's body length from its anterior to posterior end, and body width at the midpoint of its body. To obtain landmark coordinate data from the anchors, we used TPSDIG2 (*Rohlf, 2013*; *Rohlf, 2015*). Eleven landmarks (LM; six Type I, five Type III) were placed sequentially on the right and left ventral and dorsal anchors of each specimen (Fig. 1).

The set of all 11 landmark coordinates makes up a specimen's landmark configuration. Six of the landmarks are Type I (LM1, LM2, LM3, LM5, LM7, LM8), while the remainder (LM4, LM6, LM9, LM10, LM11) are Type III (i.e., semi-landmarks). LM1 and LM3 are the inner root point and the outer root point, respectively. Sandwiched between them is LM2, the groove point. LM5 is the dent point, while LM7 is the curve point. The tip point is represented by LM8. The semi-landmarks were defined relative to Type I landmarks. The horizontal (towards the outer root point) and vertical projections (towards the curve point) from LM2 intersect with the anchor outline to give LM4 and LM6, respectively. LM9 and LM10 are the intersection points between the vertical projection from LM7 and LM1 with anchor outline, respectively. The projection from LM2 perpendicular to the vertical projection from LM1 touches the anchor outline to define LM11. We used the set of landmarks LM1–LM4 and LM11 to represent the shape of the root compartment, and the set LM5–LM10 to represent the point compartment. For geometric morphometric analysis, semi-landmarks were not specially treated (e.g., employing sliding landmark analysis), following *MacLeod (2013)* that such treatments may introduce distortions to the original geometrical relationship that lead to complicated interpretations of the result. The anchor images and their corresponding landmark coordinate data have been deposited into the Data Dryad Repository (available at http://dx.doi.org/10.5061/dryad.50sg7).

## Data processing and analysis tools

We created a new R (Version 3.2.1; *R Core Team, 2015*) package called `monogeneaGM` (*Khang, 2015*) to process raw landmark coordinate data and integrate new methodological
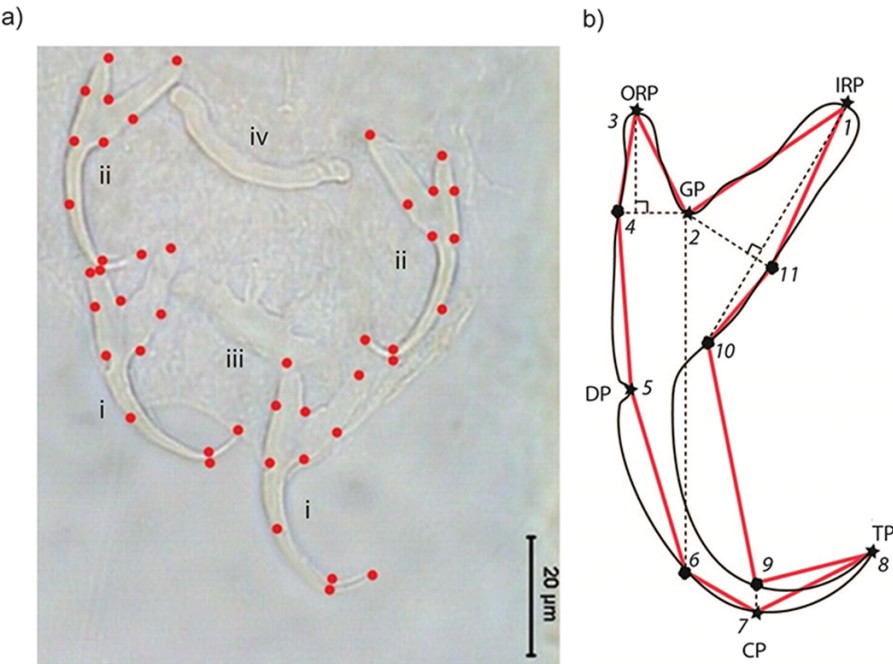

**Figure 1** (A) Landmarks of the (i) ventral and (ii) dorsal anchors of a *L. navjotsodhii* sample, digitized using TPSDIG2 (Version 2.17). The (iii) ventral and (iv) dorsal bars can also be seen in the image. (B) Landmark positions on an anchor. Type I landmarks (LM1, LM2, LM3, LM5, LM7, LM8) are indicated by stars, while Type III landmarks (LM4, LM6, LM9, LM10, LM11) are indicated by solid circles. Abbreviations: ORP, outer root point; GP, groove point; IRP, inner root point; DP, dent point; CP, curve point; TP, tip point.

developments in the current study with numerous data processing and analysis tools in R packages, such as geomorph (*Adams & Otarola-Castillo, 2013*), phytools (*Revell, 2012*), circular (*Agostinelli & Lund, 2013*), gplots (*Warnes et al., 2014*), ape (*Paradis, Claude & Strimmer, 2004*), rgl (*Adler et al., 2014*) and cluster (*Maechler et al., 2015*).

The monogeneaGM package contains a suite of R functions for three primary analyses: anchor landmark coordinate data quality control, visual checks and Generalized Procrustes Analysis; hierarchical clustering and principal component analysis; and analysis of anchor shape change using directional statistics. Some of the functions are suitable for general use. For example, the phylomorphospace visualization functions— tpColorPlot2d for two-dimensional data, and tpColorPlot3d for three-dimensional data, can take in two additional arguments: a color transparency control, which is useful for improving graphical presentation when there are substantial overlaps of data points, and an option to superimpose a user-supplied phylogeny onto defined centroids in the scatter plot, if this information is available.

## Data quality control

Despite careful slide preparation, it is inevitable that anchor images of some specimens would contain substantial amount of non-biological shape variation caused by incongruent image and object planes (*Arnqvist & Mårtensson, 1998*). The inclusion of these poor quality data in downstream analyses is undesirable, as they introduce noise into an analysis that

can potentially complicate the interpretation of results. To mitigate this problem, we developed a quality control procedure to filter out poor quality images. In this procedure, we first computed all pairwise Euclidean distances between landmarks for the left and right forms of dorsal and ventral anchors. If both left and right forms have congruent image and object planes, then by symmetry, their residual—the difference of their pairwise Euclidean distances for each landmark ($M$), should be close to zero, thus yielding a small sum of squared residuals ($M^2$). Moreover, we expect $M$ to be randomly distributed with zero mean across all average pairwise Euclidean distances ($A$) between the left and right forms. The slope of the regression equation of $M$ against $A$ ($b$) allows us to measure how well this expectation is satisfied. To be comparable with the sum of squared residuals, we squared the estimated regression slope ($b^2$), and then scaled it to be on the same order of magnitude as $M^2$. Thus, a good quality specimen would have small sum of $M^2$ and $b^2$, and vice versa. We defined the quality score $Q$, as

$$Q = 100 \times 10^{\frac{-\sqrt{M^2 + b^2}}{10}}.$$

The magnitude of this measure is straightforward to interpret—it is high (maximum 100) for good quality specimens and low (minimum 0) for poor ones. Figure 2 shows examples of poor and good quality specimens together with their Tukey Mean-Difference (TMD) plots, respectively. Specimens with $Q$ of 10 or more ($n = 437$; Table S3 and Fig. S2) were used for subsequent analyses.

## Converting pairwise euclidean distances in arbitrary units to physical units

We used a subset ($n = 96$) of the total specimens with quality score above 10 ($n = 437$) and measured the physical distances from LM1 to LM3 and from LM1 to LM5 in these samples using QWin plus image analysis software (Leica Microsystems, Germany). We then regressed the physical distances against the computed pairwise Euclidean distances to determine the linear equation for converting arbitrary distance units into their physical units (in μm). Thus, all pairwise Euclidean distances computed from raw landmark coordinates could be converted to physical distances by multiplication with a factor of 0.2 followed by addition of 0.9 (Fig. S3).

## Geometric morphometric analysis

For each species, we performed Generalized Procrustes Analysis (GPA; *Gower, 1975*; *Rohlf & Slice, 1990*) to align the sample landmark configurations for both ventral and dorsal anchors, using the gpagen function in the geomorph package (Version 2.1.1; *Adams & Otarola-Castillo, 2013*). The resulting GPA coordinates of the left and right forms were then averaged. GPA removes the effects of translation, rotation and scaling so that the resulting landmark configurations have minimum sum of squared distances with respect to the mean landmark configuration (*Adams, Rohlf & Slice, 2004*). Nevertheless, even after GPA, comparison of anchor shape variation can still be potentially confounded by the presence of non-biological variation in the landmark configuration. Specifically, if many samples of a species have anchors lying in one particular position, it would not be clear

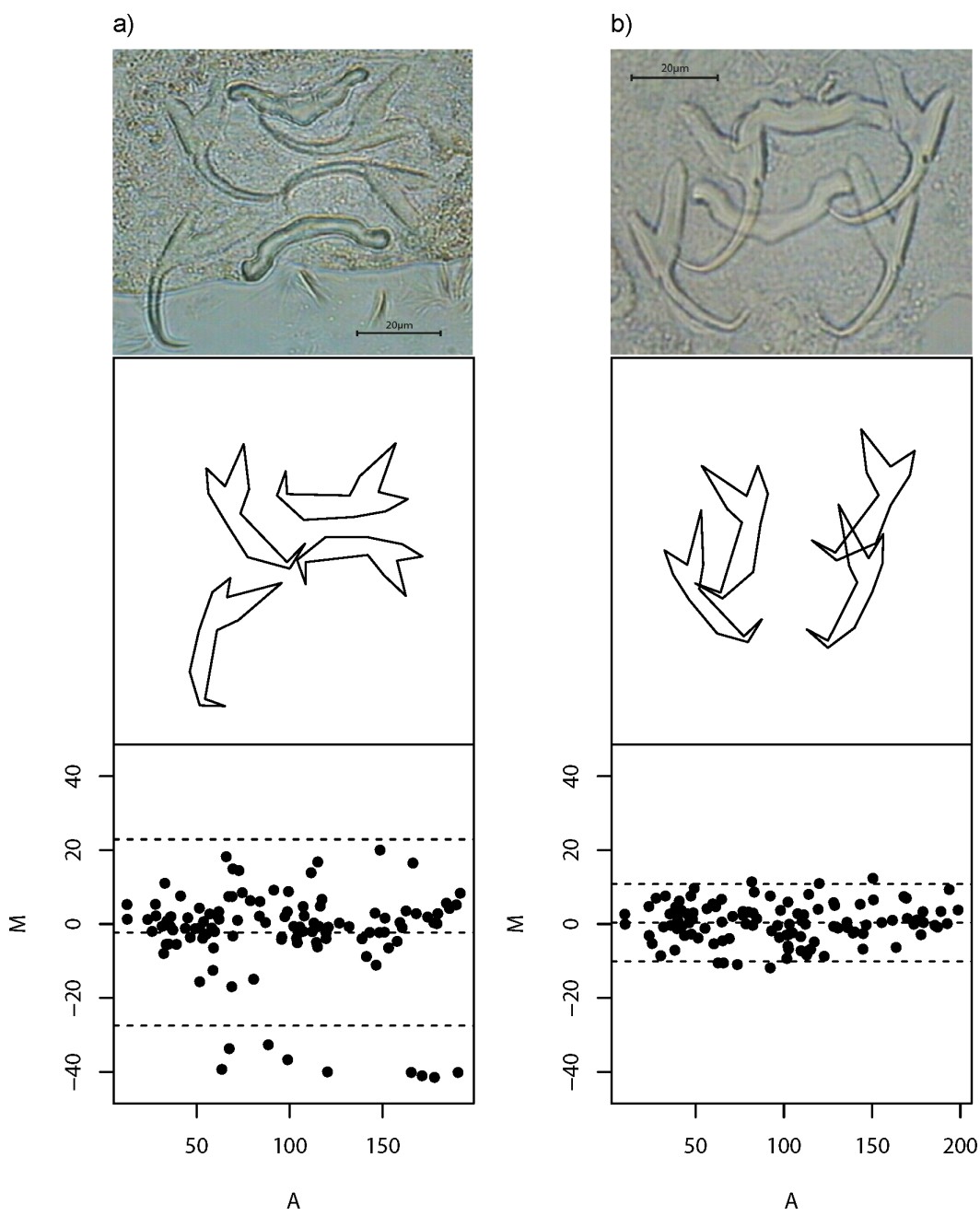

**Figure 2 Wireframe plots of anchors of *L. navjotsodhii* lying in their natural positions in the mounted slide.** (A) Example of a poor quality specimen ($Q = 5$); note the larger than expected variation between shape of (larger) left and right forms of the dorsal anchors, which shows up in a Tukey Mean Difference (TMD) plot that has relatively wide width of 95% limits of agreement (upper and lower dashed lines) as well as a fanning pattern as $A$ becomes larger. (B) Example of a good quality specimen ($Q = 30$); note the lack of shape variation between left and right forms of ventral and dorsal anchors, which shows up in a TMD plot with much narrower width of 95% limits of agreement as well as a more or less random deviation of $M$ about 0 independent of $A$.

whether variation between its members' mean GPA landmark configuration and those of other species constitutes genuine biological variation or mathematical artifact. Typical application of geometric morphometrics in non-microscopic objects (fly wings, skulls, etc.) does not usually suffer from this problem, since specimens from species to be compared can be manipulated into standardized positions before imaging.

To ensure that the landmark configurations of all 13 species were comparable, we determined the angular deviation of LM7 from the $x = 0$ line, and rotated all landmark coordinates by this amount with the origin as pivot. This has the effect of creating standardized landmark configurations for specimens across all species since the x-coordinate of LM7 is always zero after adjustment. The GPA coordinate data of all specimens thus obtained were subjected to another iteration of GPA to produce the final shape alignment, which was organized into a data matrix with rows representing specimens and columns representing the 44 GPA landmark coordinates.

## Molecular phylogenetic analysis

We used DNA sequence data from three nuclear markers: 28S rRNA, 18S rRNA and ITS1 from the 13 *Ligophorus* species to infer their phylogenetic tree. These markers are a mainstay of parasitic platyhelminth molecular phylogenetics (*Littlewood, 2008*), and the combination of fast (ITS-1) and slow (28S rRNA, 18S rRNA) evolving sequences should provide sufficient resolution for the inferred phylogenetic tree (*Lockyer, Olson & Littlewood, 2003*; *Blair, 2006*; *Waeschenbach et al., 2007*), allowing the relationship of even closely related monogenean species to be resolved (e.g., in *Gyrodactylus* (*Gilmore et al., 2012*)). Ideally, the inclusion of mitochondrial markers such as cytochrome oxidase I (COI) would provide a more representative sampling of the genome, and hence, a more reliable phylogeny. However, the absence of mitochondrial data may not impact the quality of the inferred phylogeny, since, for monogeneans, it has been shown in Diplectanidae and *Gyrodactylus* that both rRNA and mitochondrial markers are effective for species identification (*Vanhove et al., 2013*). Regardless of whether nuclear or mitochondrial genes are used, it is important to keep in mind that gene trees do not always reflect the species tree (*Maddison, 1997*).

Partial 28S rRNA (∼800 bp) and ITS1 (∼750 bp) sequence data were obtained from *Soo, Tan & Lim (2015)*, whereas 18S rRNA sequence data were generated in the present study. Briefly, the *Ligophorus* specimens were removed from host gills, identified morphologically and then preserved in 75% ethanol. Genomic DNA was extracted from samples using DNAEasy extraction kit (QIAGEN, Hilden, Germany). About 5 µl of the extracted DNA was used as template in the PCR reaction to amplify the partial 18S rRNA sequence using two primers: WormA (5′-GCGAATGGCTCATTAAATCAG-3′) (*Littlewood & Olson, 2001*) and new930F (5′- CCTATTCCATTATTCCATGC-3′) (modified from *Littlewood & Olson, 2001*). The PCR reaction (50 µl) was carried out in a solution containing 1.5 mM MgCl₂, PCR buffer (Fermentas), 200 µM of each deoxyribonucleotide triphosphate, 1.0 µM of each PCR primer and 1U of Taq polymerase (Fermentas), in a thermocycler (Eppendorf Mastercycler) using the following conditions: initial denaturation at 95 °C for 4 min, followed by 35 cycles of 95 °C, 52 °C and 72 °C for one minute each, with final

**Table 1 GenBank accession numbers of 28S rRNA, 18S rRNA and ITS1 sequences of the 13 *Ligophorus* species, with information about the latters' host species and collection location.** Sequences obtained in present study are marked with an asterisk.

| *Ligophorus* species | Host species | Locality (Malaysia) | GenBank Accession no. | | |
|---|---|---|---|---|---|
| | | | 28S rRNA | 18S rRNA | ITS1 |
| L. bantingensis | Liza subviridis | Carey Island, Selangor | KM221909 | KM221934 * | KM221922 |
| L. belanaki | | | KM221910 | KM221935 * | KM221923 |
| L. careyensis | | | KM221911 | KM221936 * | KM221924 |
| L. chelatus | | | KM221912 | KM221937 * | KM221925 |
| L. funnelus | | | KM221914 | KM262663 * | KM262662 |
| L. navjotsodhii | | | KM221920 | KM221944 * | KM221932 |
| L. parvicopulatrix | | | KM221921 | KM221945 * | KM221933 |
| L. fenestrum | Moolgarda buchanani | Langkawi Island, Kedah | KM221913 | KM221938 * | KM221926 |
| L. kedahensis | | | KM221917 | KM221941 * | KM221929 |
| L. kederai | | | KM221918 | KM221942 * | KM221930 |
| L. grandis | | Straits of Johor | KM221915 | KM221939 * | KM221927 |
| L. johorensis | | | KM221916 | KM221940 * | KM221928 |
| L. liewi | | | KM221919 | KM221943 * | KM221931 |

extension at 72 °C for 10 min. An aliquot (10 µl) from the amplicons were electrophoresed in 1.3% agarose gel, stained with ethidium bromide and viewed under an ultraviolet illuminator. The remaining 40 µl of each amplicon was purified using a DNA purification kit (QIAGEN, Hilden, Germany) and subjected to automated DNA sequencing (ABI 3730 DNA Sequencer, First Base Laboratories, Kuala Lumpur) using the same primers used for PCR amplification. Approximately 750 bp of the 18S rRNA sequence were amplified and sequenced for the 13 *Ligophorus* species (Table 1).

For phylogenetic analysis, we first aligned sequences for each marker using MAFFT (Version 7 at http://mafft.cbrc.jp/alignment/server/; *Katoh & Standley, 2013*). The alignment parameters used were the Q-INS-i iterative refinement method, and the 1PAM/κ = 2 nucleotide scoring matrix with a gap opening penalty of 1.53. We then concatenated multiple sequence alignments of the three nuclear markers (18S rRNA-ITS1-28s rRNA). The resulting supermatrix was used as input in IQ-TREE (*Nguyen et al., 2015*) to construct the maximum likelihood (ML) phylogenetic tree (*Felsenstein, 1981*; *Felsenstein, 2003*). IQ-TREE is a state-of-the-art ML tree construction pipeline that integrates DNA model selection, maximum likelihood tree search, and rapid bootstrap analysis (*Minh, Nguyen & Von Haeseler, 2013*). For model selection, we used the Bayesian Information Criteria, and did not consider "G + I" models, following *Yang (2006)* that modelling proportion of invariable sites in the presence of gamma rate variation across sites creates model identifiability issues. MEGA (Version 6; *Tamura et al., 2013*) was used to edit the resulting phylogenetic tree. We annotated the tree with the morphology of anchors, bars and male copulatory organ to allow visual assessment of overall phylogenetic and phenotypic correlation.

## Species discrimination

Discriminating a monogenean species is a complex art that involves the comparison of qualitative features of numerous anatomical structures: the male copulatory organ, female reproductive organ, anchors, bars and marginal hooks. Among the sclerotized hard parts, multivariate morphometric analyses of shape and size variables of suitable anatomical structures provide a quantitative means for species discrimination, which is invaluable for complementing the results from qualitative morphological analyses.

To visualize species clustering in low dimension morphospace, we applied Principal Component Analysis (PCA) separately for the ventral and dorsal anchors using their GPA coordinate data. The trade-off between loss of information through dimensional reduction and gain of interpretation via visualization in PCA can, however, make it difficult to judge how well members of the same species cluster together in the PCA scatter plots, especially when there are overlaps between different species clusters. To overcome this problem, we complemented PCA results with the cluster heat map (*Wilkinson & Friendly, 2008*), a powerful method for organizing high-dimensional multivariate data that allows visual detection of patterns of variation. The cluster heat map first maps numerical information in the cells of the input data matrix to corresponding color codes. Then, a hierarchical clustering algorithm is applied to cluster the samples by similarity, in such a way that within cluster variation is always smaller than between cluster variation. For the current analysis, we estimated similarity between each pair of samples using the Manhattan distance metric. The resulting distance matrix was then used as input for hierarchical clustering of samples using the Ward algorithm.

To assess the impact of applying the quality control procedure, we compared cluster heat maps generated using all samples, and using only samples that passed data quality control. Heat map construction was done using the `heatmap.2` function in the `gplots` package (Version 2.13.0; *Warnes et al., 2014*). We found the simple heat map a good alternative to inspection of the PC loadings table when trying to interpret the first few PC axes biologically.

## Testing for the presence of phylogenetic signal in anchor shape

Species with different shapes are localized in particular regions of the morphospace. When a phylogeny is superimposed onto this morphospace, a phylomorphospace is induced, and it becomes possible to evaluate whether common descent or convergent evolution is likely to have shaped phenotypic similarity (*Klingenberg & Ekau, 1996*; *Sidlauskas, 2008*; *Revell, 2014*). If anchor shape contains substantial phylogenetic signal, then we expect the phylogeny to have non-random branching patterns in phylomorphospace. Graphically, we may visualize the latter by superimposing the molecular phylogeny of *Ligophorus* on the PCA plots of the first three principal components for the ventral and dorsal anchors. Estimation of ancestral node positions in the phylomorphospace was done using the maximum likelihood method as implemented in the `fastAnc` function of the `phytools` package (Version 0.4-21; *Revell, 2012*).

We formally tested the presence of phylogenetic signal in anchor shape by applying Adams's multivariate $K_{\text{mult}}$ test (*Adams, 2014a*), implemented using the `physignal`

function (10,000 iterations) in the geomorph package (Version 2.1.1; *Adams & Otarola-Castillo, 2013*). The $K_{mult}$ statistic is a multivariate generalization of Blomberg's $K$ statistic (*Blomberg, Garland & Ives, 2003*). The phenotype (a continuous trait) of interest in the lineages of a given phylogeny is assumed to evolve in phylomorphospace according to Brownian motion. In the absence of phylogenetic signal, $K_{mult} = 0$ (i.e., phenotypic variation is independent of lineages). At $K_{mult} = 1$, phenotypic variation between taxa in the same lineage conforms to expectation under Brownian motion evolution. Values of $K_{mult} < 1$ correspond to phenotypic variation that is larger than expected between taxa of the same lineage, and values of $K_{mult} > 1$ correspond to phenotypic variation that is smaller than expected between taxa of the same lineage (*Adams, 2014a*). Statistical significance is determined via a permutation procedure under assumption of phylogenetic signal absence.

## Analysis of anchor shape and size evolution

In studying anchor shape and size evolution, we were primarily concerned with trends occurring in different clades of the phylogeny of the 13 *Ligophorus* species. To control for the effect of body size in subsequent phylogenetic regression analysis of anchor shape against anchor size, it was necessary to first test for collinearity of body size and anchor size (*Mundry, 2014*). Since body size was prone to distortion during fixation, we used the median of body length and body width of each species to reduce the impact of outliers. For analysis, the logarithm (base 10) of the product of median body length and median body width was used.

The GPA landmark coordinates of the ancestral anchor were estimated using the maximum likelihood method as implemented in fastAnc function from the phytools package (Version 0.4-21; *Revell, 2012*). Anchor shape change associated with a clade is statistically supported if mean directional change deviates significantly from uniformity in a set of landmarks. We visualized directional deviation in the 11 landmarks of both ventral and dorsal anchors using circular plots (*Agostinelli & Lund, 2013*; implemented in the circular package, Version 0.4-7). We then performed Rayleigh's test (*Batschelet, 1981*) to test for evidence against directional uniformity in each landmark. The strength of statistical evidence against mean directional uniformity in each landmark was assessed using *p*-value. Wireframe-lollipop plots (*Klingenberg, 2013*) were used to graphically summarize the mean change in direction and mean magnitude of landmark displacement from root ancestor landmark configuration.

For investigating trends in anchor size evolution, we first computed all possible pairwise Euclidean distances between the raw landmarks in each sample. Each dorsal and ventral anchor has 11 landmarks, thus generating 55 possible pairwise Euclidean distances which we used as size variables. When the loadings and variables of the first principal component (PC1) have the same sign, PC1 can be interpreted naturally as a measure of size (*Jolliffe, 2002*). Subsequently, we performed continuous character mapping (implemented using contMap function in the phytools package) of mean PC1 of the size variables of each species for ventral and dorsal anchors onto the phylogeny of 13 *Ligophorus* species to assess clade-specific patterns of anchor size evolution. The Adams-Collyer phylogenetic regression for shape response variable (*Adams, 2014b*; *Collyer, Sekora & Adams, 2015*; implemented

in the geomorph package, Version 2.1.1) was used to formally test evolutionary correlation between anchor shape and two covariates: the logarithm of body size and the logarithm of anchor size. The interaction between the two covariates was incorporated into the phylogenetic regression model if covariate collinearity could be ruled out using the Ho-Ané phylogenetic regression (*Ho & Ané, 2014*; implemented in the phylolm package, Version 2.2) under a Brownian motion model for the phylogenetic covariance matrix. For both regression analyses, *p*-values were computed via a resampling procedure with 10,000 iterations.

## Covariation of anchor shape and size with male copulatory organ morphology

*Rohde & Hobbs (1986)* hypothesized that the reproductive barrier among congeneric species that share the same host can be maintained in monogenean parasites by their having different copulatory organ morphology when attachment organs are similar (thus occupying similar microhabitats); conversely, when parasites have dissimilar attachment organs (thus occupying different microhabitats), the morphology of their copulatory organs would not show important differences, since the lack of proximity puts less evolutionary pressure on the parasites to evolve morphologically different copulatory organ. Qualitative evidence with limited number of congeneric species (*Lambert & Maillard, 1975*; *Roubal, 1982*; *Rohde et al., 1994*) supported the hypothesis's feasibility, as well as later studies with more species (*Šimková & Morand, 2008*; *Šimková & Morand, 2015*). Quantitative evaluations using larger species assemblage that relied on traditional morphometric data are available, but the interpretation of their results in support of the hypothesis was obscured by either the problem of using inflated degrees of freedom in regression analysis (e.g., *Šimková et al., 2002*) or failure to control for the effect of phylogeny (e.g., *Jarkovský et al., 2004*). With the development of new tools for geometric morphometric and phylogenetic comparative methods, we are in a position to retest the Rohde-Hobbs hypothesis. To this end, we compared the size of the male copulatory organ (mean tube length data from *Soo & Lim, 2012*; *Soo & Lim, 2015*; *Soo, Tan & Lim, 2015*) and three of its selected morphological characters (Table 2: position of copulatory organ entrance at the main lobe of accesory piece; accesory piece of male copulatory complex; shape of accesory piece of male copulatory complex) against anchor shape and size variation. Ancestral node positions were estimated as before using the fastAnc function in the phytools package.

## Morphological integration analysis

The roots of the anchor are bases for muscle attachment. Biomechanically, force exerted through the muscles and transmitted to the point compartment controls the anchor's grip strength on the gills. Because of this, we may expect the anchor to function as a single, fully integrated module (*Klingenberg, 2008*) on a priori grounds. Anchor shape is strongly constrained by either phylogeny or convergent evolution. If the latter's effect was weak, then suitable morphological characters based on variation in anchor shape can be expected to be systematically useful.

To date, only few morphological integration analyses in monogeneans have been done. Using published morphological drawings, *Vignon, Pariselle & Vanhove (2011)* investigated

**Table 2   List of morphological characters used to construct maximum parsimony trees.** All characters in Set B were taken from *Sarabeev & Desdevises (2014)*. Characters 1–6 of Set B are the same as Set A's; Characters 7–12 of Set A were constructed in the present study based on results of geometric morphometric analysis.

| Characters | Character states | Set A Included in study | Set A Index | Set B Included in study | Set B Index |
|---|---|---|---|---|---|
| **Male copulatory organ:** | | | | | |
| Position of copulatory organ entrance at the main lobe of accessory piece | (0) proximal; (1) distal; (2) medial | ✓ | 1 | ✓ | 1 |
| Accessory piece of male copulatory complex | (0) consists of two lobes (main and secondary lobes or proximal and distal ones); (1) consists of one lobe | ✓ | 2 | ✓ | 2 |
| Shape of accessory piece of male copulatory complex | (0) beak or hook-shaped; (1) claw-shaped pincer-like; (2) cross-shaped; (3) funnel-shaped; (4) open, grooved tube (rod-like) | ✓ | 3 | ✓ | 3 |
| **Female reproductive system** | | | | | |
| Vaginal canal sclerotization | (0) present; (1) absent | ✓ | 4 | ✓ | 4 |
| Distal end of sclerotized vagina | (0) funnel-shaped thin-walled; (1) funnel-shaped thick-walled; (2) scyphoid narrow; (3) scyphoid broad; (4) not observed | ✓ | 5 | ✓ | 5 |
| **Bars** | | | | | |
| Relative size of ventral and dorsal bar | (0) subequal; (1) dorsal bar longer than ventral one; (2) ventral bar longer than dorsal one; (3) not applicable | ✓ | 6 | ✓ | 6 |
| **Anchors** | | | | | |
| Ratio of shaft to point of ventral anchor | (0) less than 1.4; (1) 1.4–2.6; (2) greater than 2.6 | X | – | ✓ | 7 |
| Ratio of shaft to point of dorsal anchor | (0) less than 1.4; (1) 1.4–2.6; (2) greater than 2.6 | X | – | ✓ | 8 |
| Length of ventral anchor point | (0) 7–12 μm; (1) less than 7 μm | X | – | ✓ | 9 |
| Length of dorsal anchor point | (0) greater than 11 μm; (1) 5–11 μm; (2) less than 5 μm | X | – | ✓ | 10 |
| Relation of outer root to point of ventral anchor | (0) outer root shorter than point; (1) outer root subequal or longer than point | X | – | ✓ | 11 |
| Relation of outer root to point of dorsal anchor | (0) outer root shorter than point; (1) outer root subequal with point; (2) outer root longer than point | X | – | ✓ | 12 |
| **New characters** | | | | | |
| Shape of ventral anchor | (0) shaft scimitar-shaped, root U-shaped (1) shaft scimitar-shaped, root V-shaped; (2) shaft sickle-shaped, root U-shaped; (3) shaft sickle-shaped, root V-shaped | ✓ | 7 | X | – |
| Shape of dorsal anchor | (0) shaft scimitar-shaped, asymmetric inner and outer roots; (1) shaft sickle-shaped, symmetric inner and outer roots; (2) shaft sickle-shaped, asymmetric inner and outer roots | ✓ | 8 | X | – |
| Ventral anchor: Length from L1 to L3 | (0) 15 μm or less; (1) greater than 15 μm | ✓ | 9 | X | – |
| Dorsal anchor: Length from L1 to L3 | (0) 15 μm or less; (1) greater than 15 μm | ✓ | 10 | X | – |
| Ventral anchor: Length from L1 to L5 | (0) Less than 15 μm; (1) 15 μm–25 μm; (2) greater than 25 μm | ✓ | 11 | X | – |
| Dorsal anchor: Length from L1 to L5 | (0) 15 μm–25 μm; (1) greater than 25 μm | ✓ | 12 | X | – |

interspecific modularity of attachment organs (marginal hooks, anchors and bars) in 66 *Cichlidogyrus* (Monogenea: Ancyrocephalidae) species. More recently, *Rodríguez-González et al. (2015a)* studied intraspecific morphological integration of the root and the point compartments of anchors in *Ligophorus cephali*, using the partial least squares method in the context of shape analysis (*Rohlf & Corti, 2000*). Here, we extended their morphological integration analysis to the interspecific level in *Ligophorus*. We applied the phylogeny-aware partial least squares method based on the evolutionary covariance matrix (*Adams & Felice, 2014*) to estimate the extent of morphological integration between the ventral and dorsal anchors, as well as that of the root compartment (L1–L4 and L11) and the point compartment (L5–L10) within and between the ventral and dorsal anchors.

### New morphological characters from morphometric variables

A continuous morphometric variable can be discretized and treated as a morphological character with two or more states for use in a cladistic analysis (*Thiele, 1993*; *Rae, 1998*; *Wiens, 2001*). In doing so, the taxonomist relies on experience and intuition to select promising morphometric variables out of a potentially large set of candidates. Unfortunately, an objective means to screen the latter is generally lacking. As a result, it is difficult to assess the level of homoplasy present in the taxonomist's candidate characters. Here, we show how comparison of patterns of shape change in different clades leads to the discovery of new morphometric variables for morphological phylogenetic analysis in *Ligophorus*. A set of 12 morphological characters defined in *Sarabeev & Desdevises (2014)* that are not invariant for the 13 *Ligophorus* species (Table 2; see Table S4 for character state matrix) was chosen. We replaced the morphological characters derived from traditional morphometric measurements of anchors with new candidates derived from geometric morphometric analysis to assess their phylogenetic informativeness. To this end, we compared how well-resolved the resulting maximum parsimony trees (using PAUP; *Swofford, 2002*) were. Tree search (initial tree obtained via stepwise addition) was performed using the heuristic search option. Branch-swapping was done using the tree bisection and reconnection algorithm. Tree reliability was assessed using 1,000 bootstrap replicates and branches were collapsed if bootstrap support was below 50%.

## RESULTS

### Molecular phylogeny

The GTR + G model was the best DNA substitution model that did not incorporate the proportion of invariable sites. After mid-point rooting, the estimated ML tree (Fig. 3; 10,000 bootstrap replicates) contained two major clades. One of them consisted of species infecting *M. buchanani* (Clade I), and the other consisted of species infecting *Liza subviridis* (Clade II). Bootstrap support was high for most internal nodes, except the most recent common ancestor node of *L. parvicopulatrix*-*L. bantingensis*, and *L. grandis*-*L. fenestrum* (between 50–60%).

### Morphometry summary statistics

Table S5 gives summary statistics of anchor size, anchor shape, body size and male copulatory organ size for the 13 *Ligophorus* species.

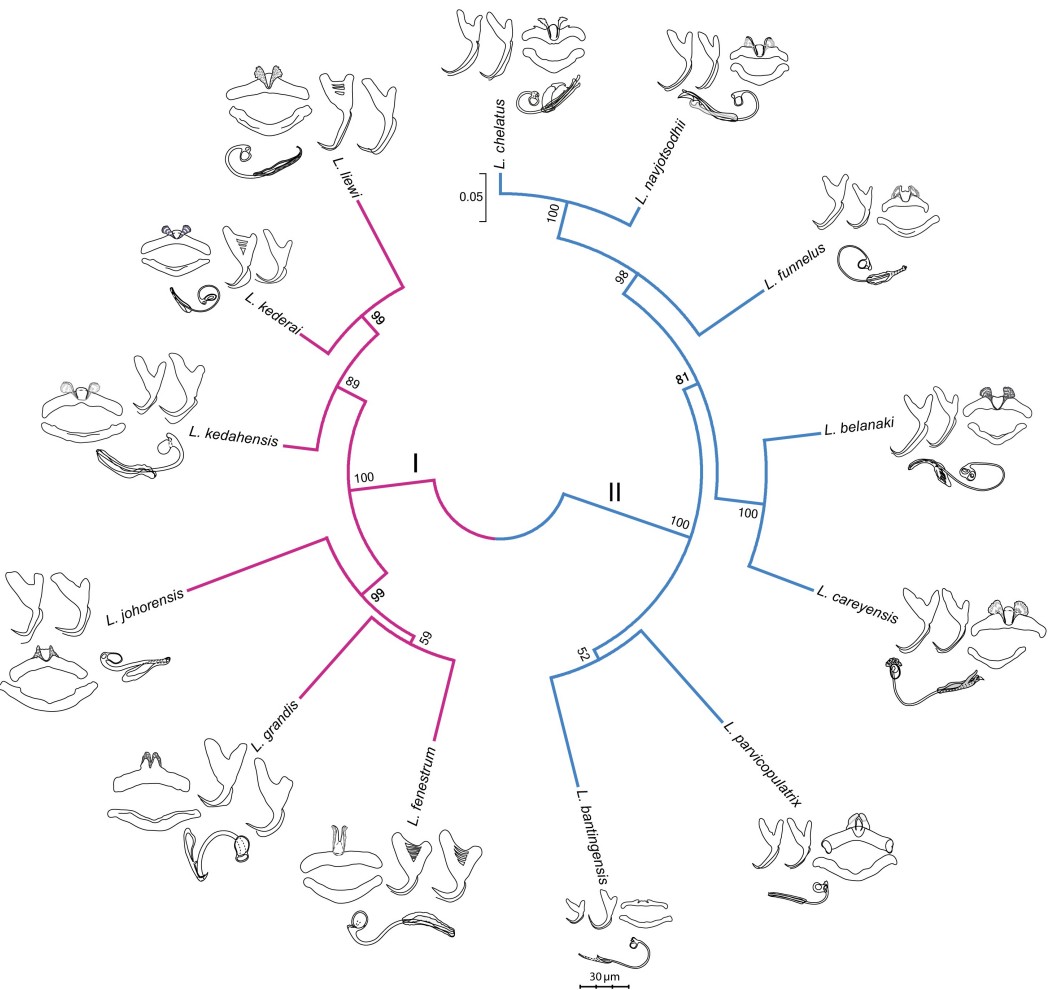

**Figure 3** **Molecular phylogeny of the 13 *Ligophorus* species inferred using the maximum likelihood method (10,000 bootstrap replicates) with annotations from three anatomical structures: anchors, bars and male copulatory organ (30% of original scale).** Species in Clade I (purple) are found in *Moolgarda buchanani*, and species in Clade II (blue) are found in *Liza subviridis*. The ventral and dorsal forms of the anchors are arranged from left to right, those of the bars from top to bottom.

## Anchor shape and phylogeny correlation

Scatter plots of GPA landmark configuration of all specimens for the ventral and dorsal anchors are given in Fig. 4 (see Figs. S4–S16 for species-specific alignments). Figure 5 shows the PCA plots of PC2 against PC1, and PC3 against PC1 for shape variables of ventral and dorsal anchors (See Fig. S17 for a three-dimensional PCA plot). The first three PC accounted for 84% and 79% of total shape variation in the ventral and dorsal anchors, respectively. To interpret these three PCs, we simultaneously compared the scatter plots of the GPA landmark configurations with the heat map of shape variable loadings (Figs. S18 and S19). Anchors with a sickle-shaped shaft had large positive values of PC1 (e.g., ventral anchors of *L. fenestrum, L. grandis, L. kedahensis, L. johorensis*), while those with a scimitar-shaped shaft (e.g., ventral anchors of *L. chelatus, L. belanaki, L. navjotsodhii*) had large negative

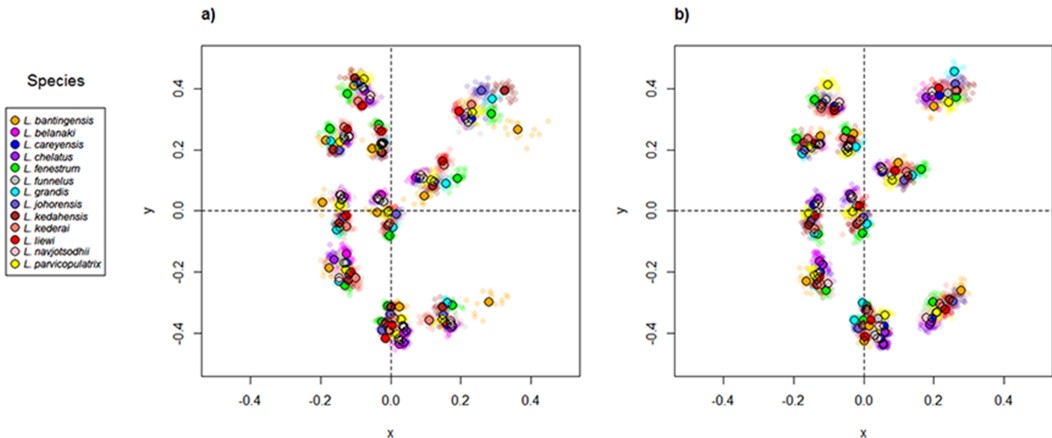

**Figure 4** **Scatter plots of Generalized Procrustes Analysis coordinates of all specimens for the (A) ventral anchors and (B) dorsal anchors.** The species centroids of each landmark are indicated by solid, colored circles.

values in ventral anchors. For ventral anchors, large positive PC2 values were associated with V-shaped root grooves (e.g., *L. bantingensis*). In contrast, the U-shaped root groove (e.g., *L. liewi*, *L. kederai*) was associated with negative PC2 values. For dorsal anchors, PC2 was positive and large for highly symmetric inner and outer roots (e.g., *L. parvicopulatrix*) and vice versa (*L. liewi*, *L. bantingensis*, *L. grandis*). PC3 did not admit a simple geometrical interpretation.

The result of Adams's $K_{mult}$ test supported the presence of significant phylogenetic signal in anchor shape ($K_{mult} = 0.948$; *p*-value $= 0.0003$). Graphically, this is reflected in the PCA plots where both clades show divergent evolutionary trajectories in phylomorphospace (Fig. 5).

## Cluster analysis of geometric morphometric data

The cluster heat map (Fig. 6) shows that variation in anchor shape alone allows the samples to be clustered unambiguously into 12 clusters corresponding to 12 of the *Ligophorus* species, confirming that between species variation is much larger than within species variation. On the other hand, with only eight specimens used, the clustering outcome was ambiguous for *L. careyensis*, whose samples were mainly clustered with *L. belanaki*.

Consistent with the detection of significant phylogenetic signal in anchor shape, hierarchical clustering revealed two major clades whose members were exactly the same as those of Clade I and Clade II. The quality of clustering using specimens that passed quality control was improved especially for species with anchors that have very similar shapes such as *L. navjotsodhii* and *L. chelatus* (misclassification error of 2.5% with filtering (Table S6); 3.8% with no filtering (Table S7)).

For each shape variable, we labelled the samples according to their membership in Clade I or Clade II, and then ranked the shape variables in descending order using the two-sample *t*-statistic to reveal inverted block structures at the top and bottom of the heat map. The shape variables that make up the top block come from the *x*-coordinates of LM2, LM4, LM7, LM9, and *y*-coordinates of LM5, LM6, LM10, their values being relatively positive in

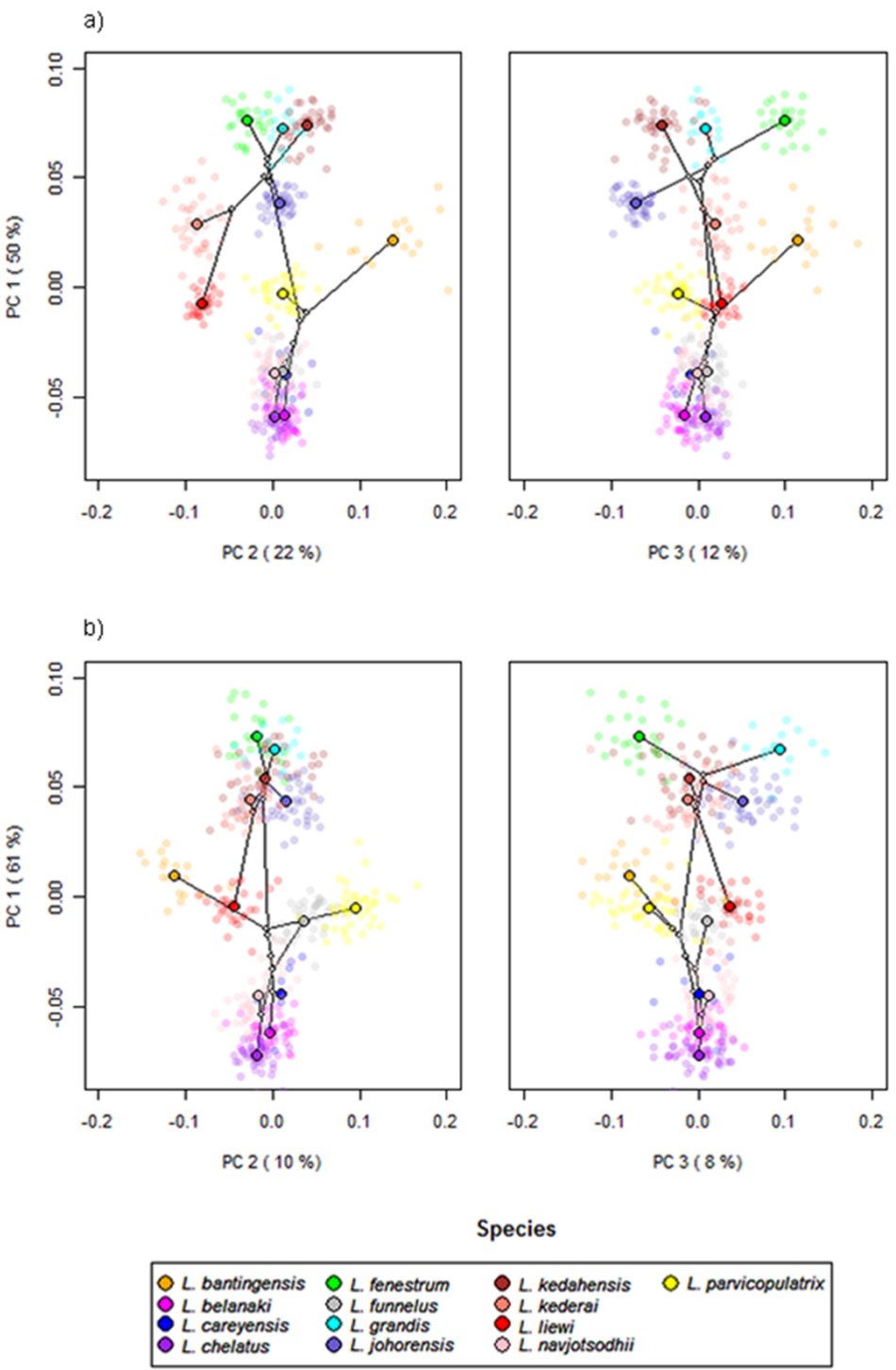

**Figure 5** **PCA plots of PC1 against PC2 and PC3 for (A) ventral and (B) dorsal anchors, with superimposed phylogeny of the 13 *Ligophorus* species.** The centroids of the species are indicated in solid colors, while individual samples are plotted in high transparency colors. The estimated principal component coordinates of the ancestral nodes are represented by small open circles.
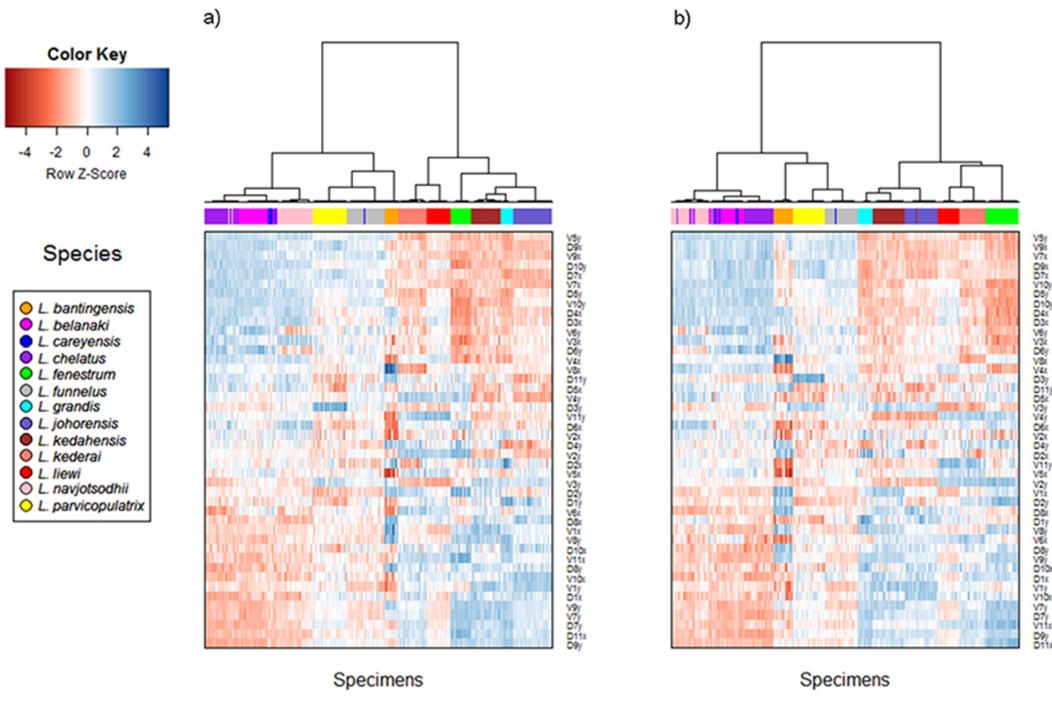

**Figure 6   Cluster heat map of specimens (column) using shape variable data (row).** The name of a shape variables consists of three parts: a prefix indicating ventral (V) or dorsal (D) anchors, a number indicating landmark index, and a suffix indicating *x* or *y* coordinate value. (A) Cluster heat map using filtered specimens with quality score of 10 or more ($n = 437$); (B) Cluster heat map using all specimens ($n = 530$).

species belonging to Clade II compared to those in Clade I. The bottom block consists of shape variables from the *x*-coordinates of LM1, LM6, LM10, LM11 and *y*-coordinates of LM1, LM7, LM8, LM9. These values were relatively negative in species belonging to Clade II compared to those in Clade I. Collectively, these variables suggest that the mean shape of the anchor shaft in Clade II was more elongated and scimitar-like (i.e., LM5, LM6, LM10 and LM7, LM8, LM9 are relatively farther from each other) while that in Clade I was more robust and sickle-like (i.e., LM5, LM6, LM10 and LM7, LM8, LM9 are relatively closer to each other).

## Anchor shape and size evolution

The wireframe-lollipop graphs (Fig. 7) show patterns of shape changes in the ventral and dorsal anchors of both clades that are consistent with those inferred from the cluster heat map and PCA. The circular plots (Figs. S20 and S21) provide more details at the level of individual landmarks. The explicit visualization of the direction and magnitude of GPA-landmark coordinate deviation relative to the ancestral form provides insights into selection of new morphometric variables suitable as morphological characters. Specifically, for two landmarks, if their mean directional change is divergent in one clade but convergent in another, then the interlandmark distance is expected to be of value for discriminating the two clades. To be easy to measure, the landmarks should be of Type I. Thus, the distance from LM1 to LM3 and from LM1 to LM5 were found to be good candidates. We found

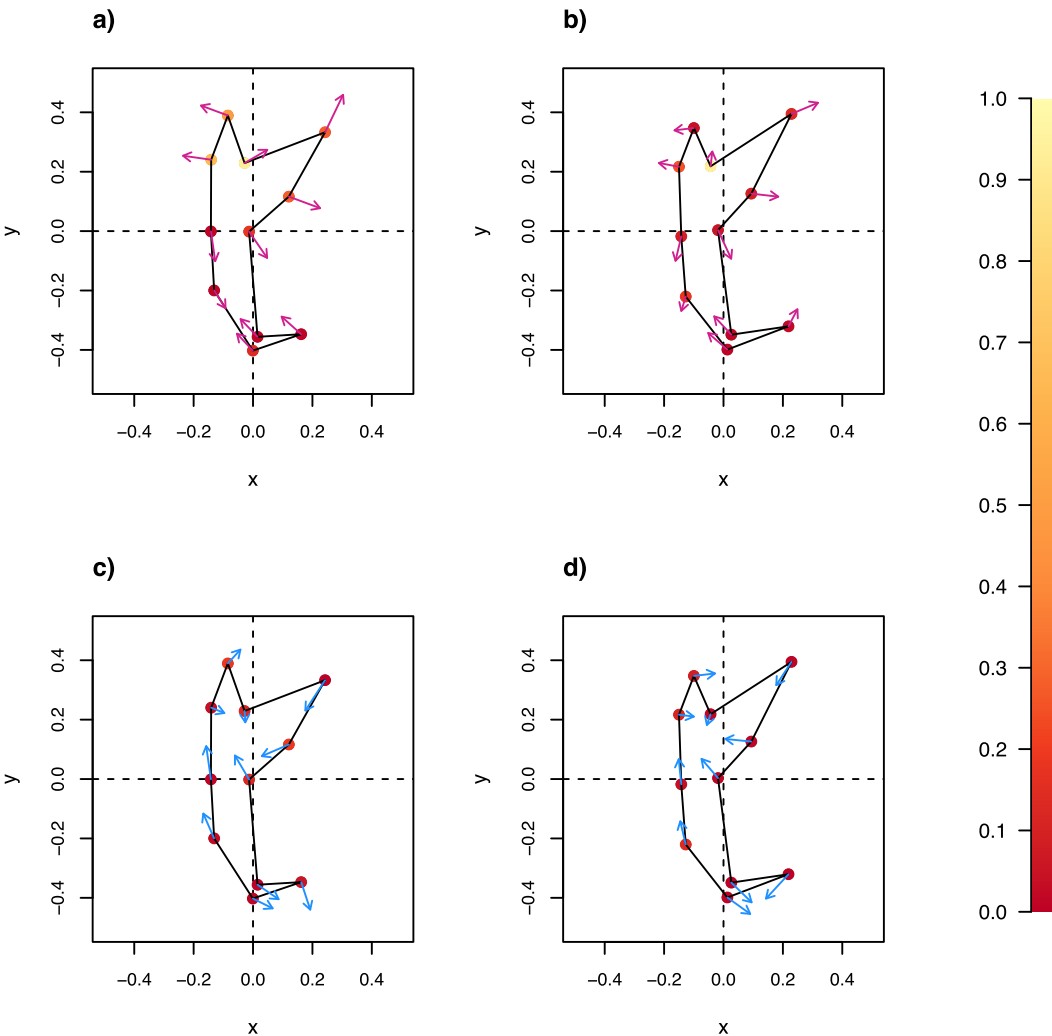

**Figure 7  Wireframe-lollipop plots of mean shape change relative to estimated ancestral mean shape in (A) ventral anchors of Clade I (purple); (B) dorsal anchors of Clade I; (C) ventral anchors of Clade II (blue); (D) dorsal anchors of Clade II.** The *p*-value of Rayleigh test for uniform direction change at each landmark is indicated as a colored solid circle. The color bar maps color tones to their corresponding *p*-values.

the common practice of using the inner root length (IL) and the outer root length (OL) (distance from LM1 to LM7 and LM3 to LM7, respectively) to be suboptimal since both LM3 and LM7 had almost parallel directional changes (Figs. 7A, 7C and 7D), whereas mean directional change in LM1 and LM7 did not show any clear patterns of divergence in one clade and convergence in another (Fig. 7) to be able to show large variation between Clade I and Clade II. Figure 8 (see also Figs. S22–S23) shows that it is possible to define cut-offs for the LM1–LM3 (15 μm) and LM1–LM5 (25 μm) distances that result in discrimination of Clade I from Clade II, but no reasonable cut-offs for IL and OL lead to similar results.

The average median body size of species in Clade I was significantly larger compared to Clade II (3.2 times; 95% confidence interval body size ratio = [1.5, 7] ). However, larger body size was not correlated with larger ventral or dorsal anchor size, after controlling for
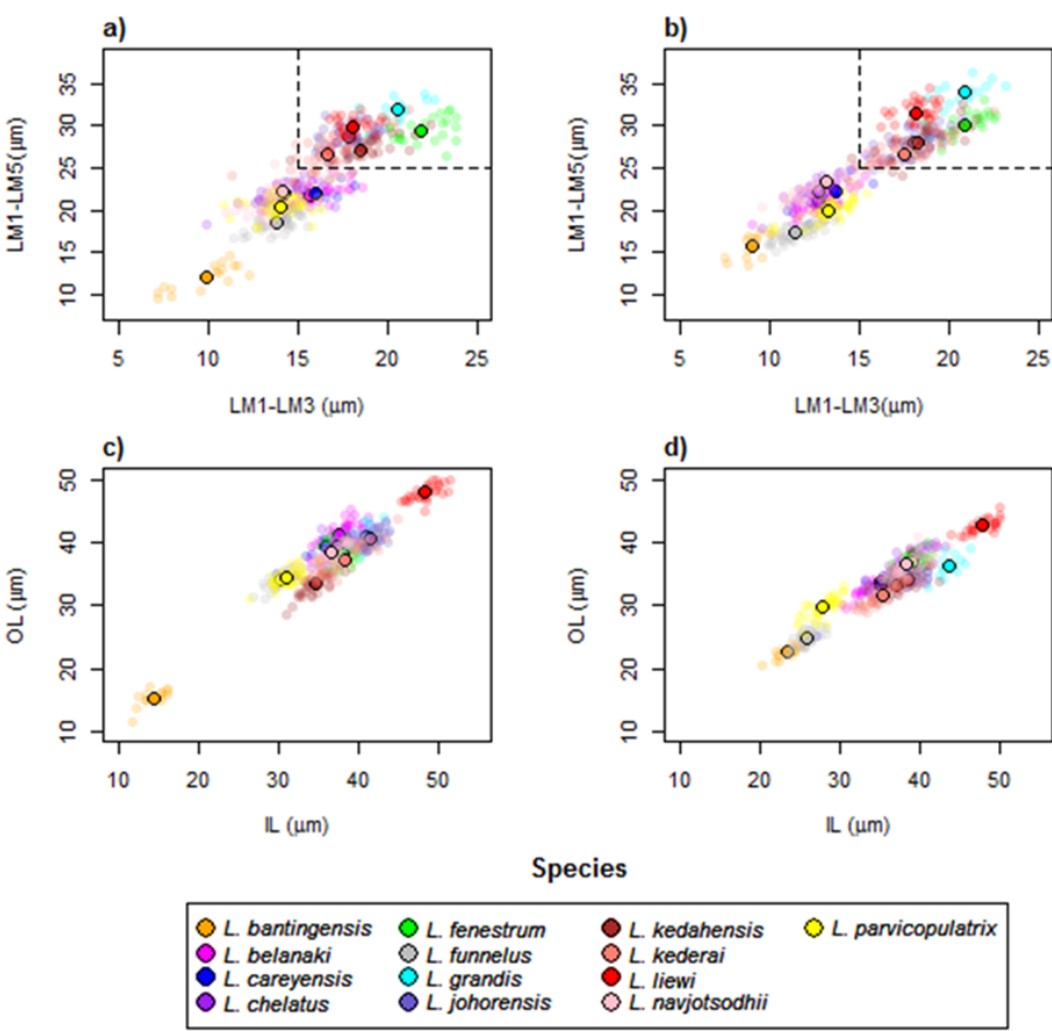

**Figure 8** **Scatter plots LM1–LM5 distance against LM1–LM3 distance for (A) ventral and (B) dorsal anchors.** The dashed lines are cut-offs on the *x* and *y* axes that allow complete discrimination of Clade I from Clade II. Scatter plots of outer length (OL) against inner length (IL) for (C) ventral and (D) dorsal anchors. No cut-offs on the *x* and *y* axes permit complete discrimination of Clade I from Clade II.

the effect of phylogeny (Ho-Ané phylogenetic regression *p*-values > 0.2). Consequently, both of them could be treated as independent covariates. The species in Clade I generally had larger anchors with sickle-shaped shaft (Fig. 9; ventral anchor mean = 170µm, standard deviation (SD) = 24 µm; dorsal anchor mean size = 165 µm, SD = 16 µm), whereas those in Clade II had smaller anchors (ventral anchor mean size = 140 µm, SD = 35 µm; dorsal anchor mean size = 133 µm, SD = 24 µm) with scimitar-shaped shaft. Size decrease was most striking in *L. bantingensis*, being 2.7 SD and 1.8 SD below the mean of all species for the ventral and dorsal anchors, respectively. Conversely, size increase was most prominent in *L. liewi*, with 1.7 SD above the mean of all species for both ventral and dorsal anchors. Nonetheless, the within clade trajectory for some species may sometimes show considerable variation from a clade's average trajectory. For example, *L. liewi* evolved a more slender

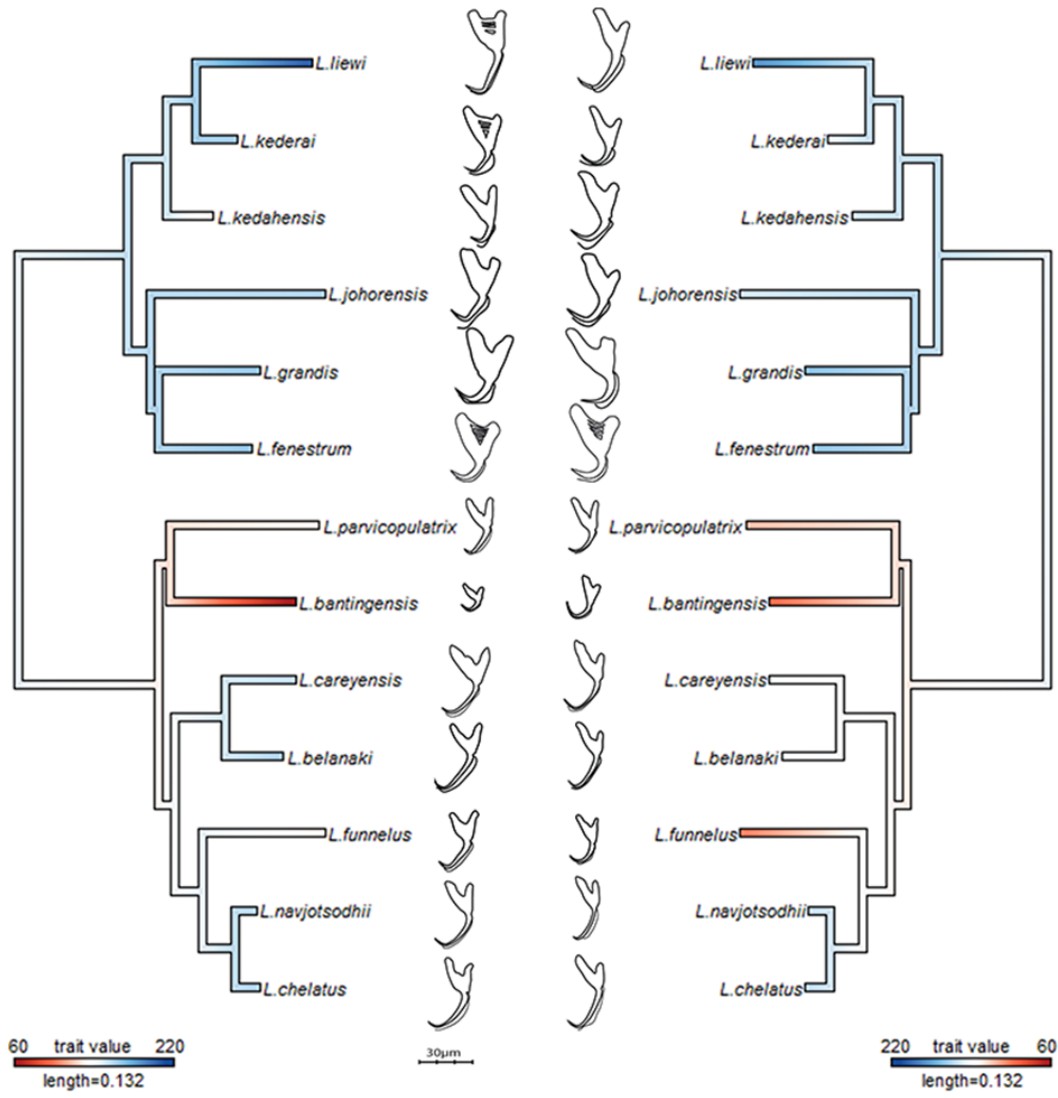

**Figure 9** Continuous character mapping of anchor size (in μm) of ventral (left) and dorsal (right) anchors onto the maximum likelihood phylogeny of the 13 *Ligophorus* species.

shaft for its ventral and dorsal anchors, which is closer to the scimitar shape found in most species in Clade II, even though its anchor size was the largest. In contrast, *L. bantingensis* evolved sickle-shaped shafts in its ventral and dorsal anchors (common in Clade I species like *L. fenestrum* and *L. grandis*) even though it had the smallest anchor size.

Results from the Adams-Collyer phylogenetic regression indicated that the interaction of body size and anchor size was not statistically significant in the dorsal anchor ($p$-value $= 0.2$), but anchor size was a significant predictor of anchor shape ($p$-value $= 0.01$). For the ventral anchor, the interaction of body size and anchor size was a significant predictor of anchor shape ($p$-value $= 0.02$). Since body size and anchor shape were not significantly correlated, we may expect similar anchor shape to be found across a range of body sizes (Fig. S24).

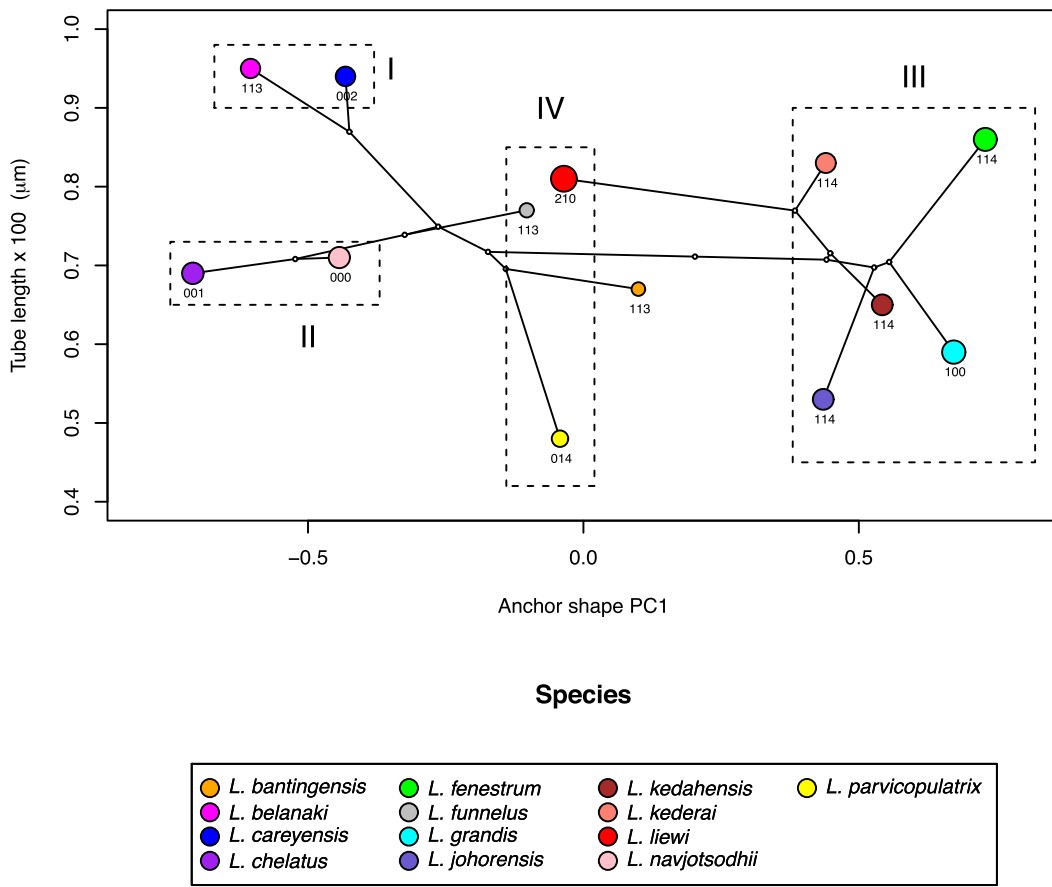

**Figure 10 Scatter plot of male copulatory organ tube length against dorsal anchor shape.** The size of dorsal anchors is proportional to the circle diameter. When anchor size and shape are similar, species that have similar size for the male copulatory organ show variation in the latter's morphology (I and II), whereas those with similar morphology of male copulatory organ show variation in the latter's size (III).

## Patterns of morphometric and morphological variation in the male copulatory organ and anchor

Where the male copulatory organ was similar in size among closely-related species with similar anchor shape and size, its morphology varied (I and II in Fig. 10). *Ligophorus belanaki*, *L. careyensis*, *L. navjotsodhii* and *L. chelatus* shared a most recent common ancestor, whose ancestral character states for the three male copulatory organ characters could be inferred as 113 on a parsimony criterion. The divergence of *L. careyensis* from *L. belanaki* did not involve major changes in anchor shape, size and size of male copulatory organ, but on the latter's morphology, which acquired three changes to become 002. Similarly, The most recent common ancestor of *L. navjotsodhii* and *L. chelatus* probably evolved character states 000 or 001 from 113, and divergence of these two species was associated with a change in the third character state, with only relatively minor change in either anchor shape and size or copulatory organ size.

In contrast, where the male copulatory organ was similar in morphology among closely-related species with similar anchor shape and size, its size varied (III in Fig. 10). *Ligophorus*

*kederai*, *L. grandis*, *L. kedahensis* and *L. johorensis* have similar anchor shape and the same character states 114 for the morphology of their male copulatory organ. Consistent with prediction from the Rohde-Hobbs hypothesis, substantial variation in the size of their male copulatory organ was observed. It is possible for size and morphological variation to co-occur in the male copulatory organ, as shown in the divergence of *L. grandis* and *L. fenestrum* from their common ancestor. Finally, species with similar anchor shape (IV in Fig. 10) showed large variation in both their male copulatory organ size and morphology.

## Morphological integration

The shapes of both ventral and dorsal anchors were strongly and significantly correlated (evolutionary correlation = 0.84, Adams-Felice test $p$-value = 0.003). Additionally, there was tight integration between the root and point compartments of the ventral (evolutionary correlation = 0.85, Adams-Felice test $p$-value = 0.004) and dorsal anchors (evolutionary correlation = 0.86, Adams-Felice test $p$-value = 0.001). Thus, the entire anchor can be considered as a single, fully integrated module. Across the ventral and dorsal anchors, integration of the point compartments was strong (evolutionary correlation = 0.92, Adams-Felice test $p$-value = 0.0001) but that of the root compartments was weaker (evolutionary correlation = 0.79, Adams-Felice test $p$-value = 0.03). Figure S25 provides a graphical summary of the results of the morphological integration analysis.

## Phylogenetically informative morphometric variables

For the current 13 *Ligophorus* species, the maximum parsimony tree estimated using the set of morphological characters containing discretized LM1–LM3 and LM1–LM5 distances and anchor shape was better resolved (Fig. 11) . Clade I and Clade II were clearly identified, and the partially resolved relationships within each clade were also congruent with those of the molecular phylogeny's. In contrast, using morphological characters of the anchors derived from traditional morphometrics as in *Sarabeev & Desdevises (2014)* produced a maximum parsimony tree that was mostly reticulate and unable to distinguish Clade I and Clade II.

# DISCUSSION

## Data quality control

We are not aware of any geometric morphometric analyses of anchors in monogeneans that currently implement specimen quality control. Specimen quality introduces an important source of non-biological variation into observed anchor shape variation, the impact of which depends on whether the data would be analyzed at the intra or interspecific level. Thus, while the inclusion of specimens that failed quality control into hierarchical clustering did not fundamentally change species discrimination outcome in this study, it is important to control for this confounder where intraspecific variation can be expected to impact conclusions of an analysis, for example, when investigating mean directional change in landmarks of anchors (Fig. 7), or testing for association between intraspecific anchor shape variation and evolutionary potential of a species (*Rodríguez-González et al., 2015a*). In the current study, we observed up to 50% loss of specimens (*L. fenestrum*) due to low quality
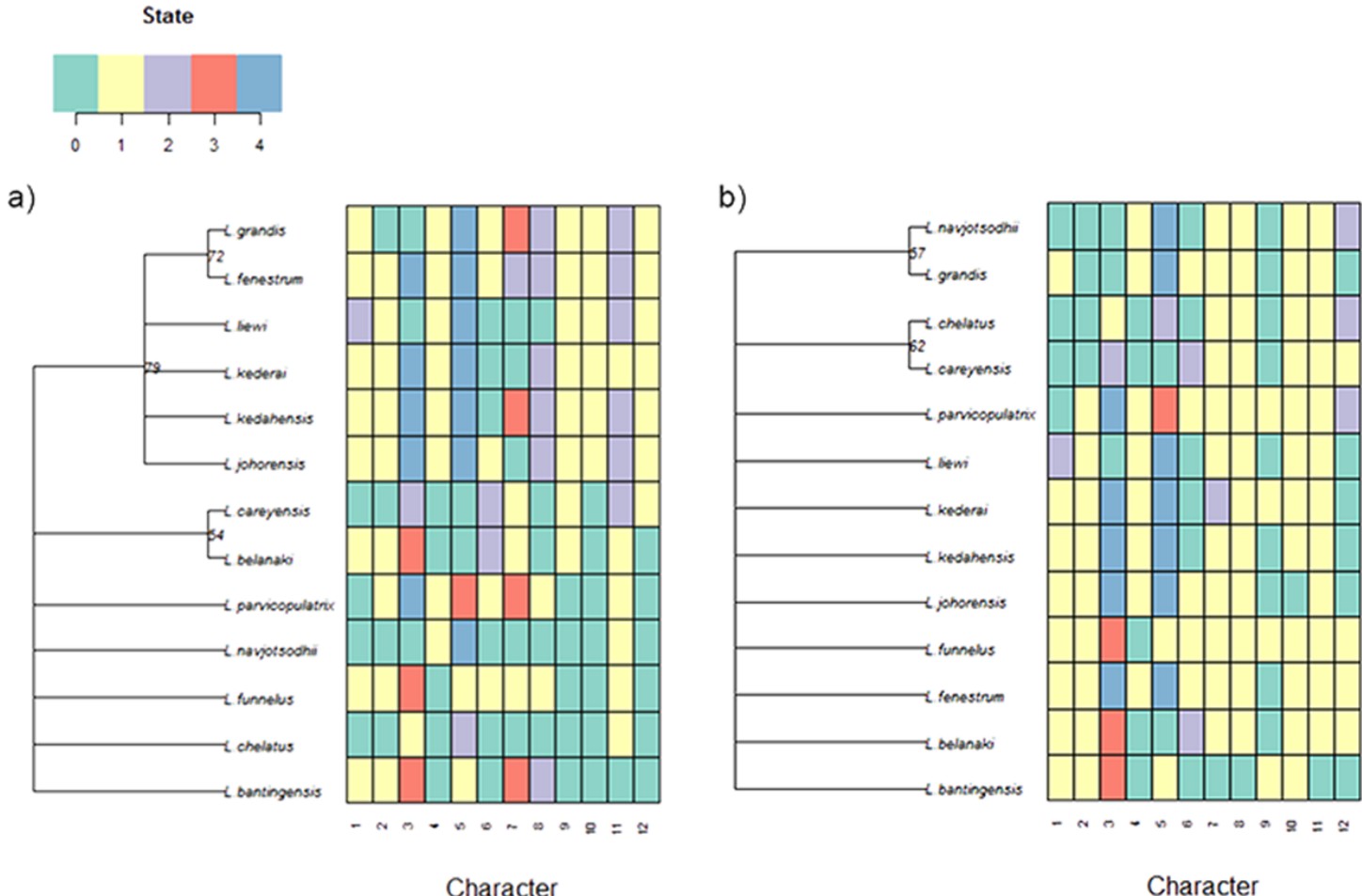

**Figure 11** **Color-coded morphological character state data for the 13 _Ligophorus_ species and their estimated maximum parsimony phylogeny (1,000 bootstrap replicates).** (A) Result using the set of morphological characters (7–12) that contain discretized LM1–LM3 and LM1–LM5 distances and anchor shaft shape. (B) Result using anchor morphological characters derived from traditional morphometrics in _Sarabeev & Desdevises (2014)_.

score. Assuming this value optimistically as an upper bound, then at least 40 specimens per species would have to be obtained in order to anticipate 20 or more specimens passing quality control. An ideal case like this may not always be possible, since sampling trips do not always yield sufficient study material.

The assumption of symmetry in the left and right side of the ventral and dorsal anchors used in the quality control procedure has a caveat that should be noted. Poor quality specimens that arise as a consequence of incongruent image and object planes are confounded with specimens that show fluctuating asymmetry. If fluctuating asymmetry (_Graham et al., 2010_) is common, then a large proportion of specimens would have been discarded using the quality control procedure, but this is not the case in the present study, where about 82% (437/530) of all specimens were retained. Careful observations over multiple well-prepared mounts should help researchers decide the appropriateness of implementing the quality control step for specimens from a particular monogenean genus.

Variation in quality scores can be attributed to several sources, such as the method of slide preparation, the quality of camera lens and software used for capturing images, and the skill and experience of the data gatherer. In this study, a single data gatherer (OYM Soo) prepared and acquired the landmark data, using the same compound microscope and computer. Because of this, we expect other factors to explain the poor quality scores. Interestingly, we note that species that had larger proportion of specimens failing quality control tend to have body and/or anchor size that were relatively large or small. In the case of *L. grandis* ($\log_{10}$ median body size 2.1 SD larger from total mean; dorsal anchor size 1.0 SD larger than total mean), *L. fenestrum* ($\log_{10}$ median body size 1.6 SD larger than total mean; dorsal anchor size 0.9 SD larger than total mean), and *L. bantingensis* ($\log_{10}$ median body size 0.9 SD smaller than total mean; ventral anchor size 2.7 SD smaller than total mean), we observed about 30%, 50% and 45% of the specimens failing quality control ($Q$-score < 10; Fig. S2), respectively. A possible explanation may be that the robust anchors of *L. fenestrum* and *L. grandis* have uneven thickness at the root and point regions, which makes them difficult to evenly flatten on slides. The large body bulk may also further hinder effective flattening. Monogenean anchor thickness is not usually measured but may be indirectly inferred through 3D-modelling (*Teo, Sarinder & Lim, 2010*; *Teo, Sarinder & Lim, 2013*). Since size and physical inertia are positvely correlated, the small body and anchor size of *L. bantingensis* make specimen orientation on the slide sensitive to variation in force applied during slide flattening.

## Phenotypic plasticity in anchor shape

Different species were found to have varying levels of intraspecific phenotypic plasticity in this study. While within species shape variation in both ventral and dorsal anchors was large in some species (*L. kedahensis*, *L. parvicopulatrix*), it was limited in others (*L. grandis*, *L. liewi*). Interestingly, the generalist *L. bantingensis*, which has been reported to be found in two small fish hosts (*Kritsky, Khamees & Ali, 2013*): *Liza abu* (body length range: 12–15.5 cm) and *Liza klunzingeri* (body length range: 14–18 cm), had the largest intraspecific shape variation in its ventral anchor, particularly in its root compartment (PC2). Phenotypic plasticity within species likely promotes divergence by increasing the adaptability to different gill microhabitats (*Rohde & Watson, 1985*; *Poisot & Desdevises, 2010*; *Pfennig et al., 2010*; *Rodríguez-González et al., 2015a*), and is generally considered to be important for generalist species (*Van Valen, 1965*; *Kaci-Chaouch, Verneau & Desdevises, 2008*; *Šimková et al., 2013*).

## Integrative geometric morphometric analysis supports the Rohde-Hobbs hypothesis

Evidence for supporting the Rohde-Hobbs hypothesis has traditionally come from integrating spatial distribution data of monogeneans on gill microhabitats (e.g., *Rohde, 1977*; *Ramasamy et al., 1985*; *Koskivaara, Valtonen & Vuori, 1992*) with morphological data of the monogenean species (e.g., Fig. 6.3 in *Šimková & Rohde, 2013*). These efforts were very laborious, but crucially established anchor shape-microhabitat association. Benefitting from such insights, our current integrative geometric morphometric analysis was able to

reveal patterns consistent with the hypothesis's predictions on how male copulatory organ size and morphology vary with respect to anchor shape (Fig. 10), despite the absence of spatial distribution data for the 13 *Ligophorus* species across gill microhabitats.

## Morphological integration, phylogenetic signal and morphological phylogenetics

In their intraspecific study of morphological integration between the root and point compartments in *L. cephali*, *Rodríguez-González et al. (2015a)* reported only modest degree of integration in the same anchor, but stronger compartmental integration between the ventral and dorsal anchors. On the other hand, using interspecific data, we demonstrated a much stronger degree of integration between the root and the point compartments within anchors, and showed relatively weaker integration of the root compartments between ventral and dorsal anchors. Intuitively, intraspecific compartmental integration within the same anchor is expected to be high, so a possible explanation for the discrepancy may be the lack of a quality control procedure for filtering poor quality slides. Without the latter, it seems difficult to rule out the possibility that the observed intraspecific anchor shape variation in *L. cephali* may contain non-trivial amount of artifactual noise.

Generally, a certain degree of homoplasy may be expected in the morphology of attachment organs in parasites, on grounds that functional requirements for attaching to the host and adapting to within-host microhabitats would override shape constraints imposed by phylogeny (*Morand et al., 2002*). Such form of adaptive evolution can cause $K_{mult}$ to become less than 1 (*Blomberg, Garland & Ives, 2003*). In the present *Ligophorus* phylogeny, the $K_{mult}$ value of 0.948 is significantly larger than 0 (*p*-value 0.0003), but slightly biased downwards from the expected value under Brownian motion evolution for anchor shape. The reason seems to be that, although anchor shape is more or less lineage-dependent in the present set of *Ligophorus* species, three species: *L. liewi* (Clade I), *L. funnelus* (Clade II) and *L. parvicopulatrix* (Clade II), have similar shaft shape (Fig. 9), despite being from different lineages, thus lowering $K_{mult}$. To put the magnitude of $K_{mult}$ in perspective, its univariate version—Blomberg's $K$ statistic, has been found to be generally less than 1 for numerous primate traits, exceeding 1 only for brain size (*Kamilar & Cooper, 2013*).

The phylogeny (*Bakke, Harris & Cable, 2002*; *Vanhove et al., 2015*), immunophysiology (*Buchmann & Lindenstrøm, 2002*), ecology (*Šimková et al., 2006*), and behaviour of the fish hosts (*Cable et al., 2002*; *Mendlová & Šimková, 2014*) all contribute to determine patterns of infection and transmission, host specificity, and subsequent mode of speciation in their monogenean parasites (*Littlewood, Rohde & Clought, 1997*; *Cribb, Chisholm & Bray, 2002*; *Whittington & Kearn, 2011*; *Vanhove & Huyse, 2015*). Therefore, it is not surprising that different monogenean families may show different degrees of morphology-phylogeny covariation. In *Lamellodiscus* (Family: Diplectanidae), the lack of cospeciation between host-parasite (*Desdevises et al., 2002*) suggests that adaptation to host after host-switching would lead to convergent evolution in some of the morphologies (but see *Machkewskyi et al., 2014*). Indeed, *Poisot, Verneau & Desdevises (2011)* found morphological features of sclerotized haptor and male copulatory organs to be weakly associated with phylogeny in
*Lamellodiscus.* On the other hand, in *Cichlidogyrus* (Family: Ancyrocephalidae), sclerite (including anchors) shape was found to contain significant phylogenetic signal (*Vignon, Pariselle & Vanhove, 2011*). However, this conclusion needs to be qualified in light of the recent demonstration of a host-switching event in *Cichlidogyrus* (*Messu Mandeng et al., 2015*), which was missed in *Vignon, Pariselle & Vanhove (2011)* because the host range (at the familial level of the fish hosts) of *Cichlidoygrus* was not taken into consideration. For *Ligophorus*, the significant phylogenetic signal found in the anchors does not appear to need similar qualification, since its host range is currently known to be restricted to mullets. In an examination of over 1,000 fish from 10 fish families in the Black Sea, *Öztürk & Özer (2014)* observed that *Ligophorus* was restricted to mullets (*Mugil cephalus* and *Liza aurata*), with almost 100% prevalence in the hosts. Nevertheless, since it has been reported that a combination of host-switching (within fish host family) and intra-host speciation events probably generated the present diversity of *Ligophorus* species in the Mediterranean basin (*Blasco-Costa, Míguez-Lozano & Balbuena, 2012*), a more definitive conclusion regarding phylogenetic signal in the anchors requires additional tests using samples of *Ligophorus* species from that region.

The anchor morphology-phylogeny correlation shown for *Ligophorus* in the present study suggests new morphological variables from the anchors that are potentially useful in morphometric analysis and also morphological phylogenetics. In particular, we suggest more active exploration of the newly proposed morphometric variables: the distance between the inner root point and the outer root point (LM1–LM3), and the distance between the inner root point and the dent point (LM1–LM5), as alternatives to existing usage of the inner root length (IL) and the outer root length (OL) in traditional morphometric analysis. The discretization of the two proposed morphometric variables: anchor shaft shape and anchor shaft shape, can also be further considered in future morphological phylogenetic analysis.

## Anchor shape and size correlation with host and ecological factors

In the present study, the modest levels of anchor shape-size covariation revealed through the Adams-Collyer phylogenetic regression analysis for the 13 *Ligophorus* species suggest that, apart from the effect of shared ancestry, anchor shape-size covariation is likely non-trivially constrained by additional factors, one of which could be their biomechanical compatibility. Another factor is host size and ecology. On average, the larger *Moolgarda buchanani* (body length range 35–48 cm) harbored larger *Ligophorus* species, whereas the smaller *Liza subviridis* (body length range 25–30 cm) harbored smaller *Ligophorus* species.

The current analysis showed that *L. bantingensis* evolved anchors of a smaller size but retained the sickle-shape shaft common in Clade I, which is associated with larger anchor size. The observed small anchor size is consistent with the hypothesis that small or medium-sized attachment organs are generally associated with a generalist lifestyle, as these sizes expand the host range of monogeneans to small or medium-sized hosts, which are generally more diversified than larger hosts (*Morand et al., 2002*).

In the present study, open seas (Langkawi Island) yielded only *M. buchanani* samples whereas sheltered marine environment yielded both *M. buchanani* and *Liza subviridis*

samples. This suggests that perhaps the larger and more robust anchors in *Ligophorus* species infecting *M.buchanani* may be a consequence of the hosts' adaptation to rough open seas, where strong water currents are expected. In such conditions, burst swimming is an important locomotory strategy for the fish host, not only to negotiate strong currents, but also to chase swift preys and avoid strong predators (*Koehn & Crook, 2013*). Indeed, for skin-parasitic fish monogeneans, their attachment appendages must be able to withstand dislodgement by strong water currents pushing against the fish surface as the fish host performs burst swimming (*Kearn, 2014*). Concurrently, it seems plausible that burst swimming may also, for a short time, cause strong water currents to flow through the gill chamber, thus imposing a selective pressure for more robust anchors in monogeneans that live on the gills.

The U-shaped root groove in some species infecting *M. buchanani* (e.g., *L. liewi, L. kederai* and *L. fenestrum*), which could result from the accretion of sclerotic material in the space between the inner and the outer root (K Rohde, pers. comm., 2014), may have substantially expanded the root base, providing space for connection with more muscle tissues. This would likely have resulted in anchors with stronger contraction strength, necessitating the evolution of the shorter but more robust sickle-shaped shaft, hence the finding of tight morphological integration between the root and point compartments.

Interestingly, U-shaped roots and sickle-shaped shaft are also common in *Cichlidogyrus* species (*Vignon, Pariselle & Vanhove, 2011*), which infect the cichlids in Africa. Ecologically, it seems unlikely that such robust shapes would have evolved in freshwater environments, where cichlids are mostly found. If the ancestors of cichlids and their monogenean parasites were marine in origin, or host switching occurred with contact between salt-tolerant cichlids and the marine ancestor of the monogenean parasite, the observed robust shapes would then be inherited, with their shapes constrained by phylogeny. While some studies (*Murray, 2001*) suggested that the ancestors of cichlids were likely marine, others were sceptical (*Chakrabarty, 2004*; *Sparks & Smith, 2005*). Joint consideration of the morphology and phylogeny of monogenean fauna of cichlids and other marine fishes is probably required (*Pariselle et al., 2011*) to assess the competing claims. For example, a recent molecular phylogenetic analysis using 28S rRNA sequence (*Tan, 2013*; Fig. S26) indicated that *Ligophorus* (marine) and *Cichlidogyrus* (mostly freshwater) species shared the most recent common ancestor.

In three species (*L. grandis, L. funnelus* and *L. liewi*), fenestration in the anchor base seems to be an invariant character state, as all examined specimens ($n = 22, 50, 32$, respectively) showed consistent presence of fenestration. Presently, the ecological significance of fenestrations in anchors is unclear. Some progress may be possible with biomechanical studies (e.g., *Wong & Gorb, 2013*) that compare whether fenestrated and non-fenestrated anchors differ significantly in their resistance against turbulence and strong water currents. The present phylogenetic analysis suggests that fenestration is not a synapomorphic character state (Fig. 3), but a clearer picture requires more extensive taxa sampling.

## Outlook for geometric morphometric analysis of sclerotized haptoral structures in monogenean parasites

The use of less sophisticated methods could have also answered the biological problems that were studied in the present work. For example, to infer the presence of phylogenetic signal in attachment organs, or to investigate anchor size correlation with parasite and host phylogeny, one could map qualitative shape and size categories of the organ of interest onto the molecular phylogeny of the parasites, as done in *Šimková et al. (2006)*. Validation of the Rohde-Hobbs hypothesis could be done by correlating parasite abundance data in different gill microhabitats with their attachment organ and male copulatory organ morphology. The lack of a unified quantitative framework, however, forces the researcher to use different data types to address each question. We have shown in the present work that, if geometric morphometric data is used as a starting point, all the preceding questions can be simultaneously answered. In fact, the geometric morphometric approach is the key that unlocks the potential of shape information in the organs of interest for broader research questions, such as species and biogeographical population discrimination (*Vignon & Sasal, 2010*), morphological integration (e.g., *Vignon, Pariselle & Vanhove, 2011*; *Rodríguez-González et al., 2015a*), and more recently, canalization and developmental stability (*Llopis-Belenguer et al., 2015*), all of which cannot be answered satisfactorily using traditional morphometric or qualitative methods.

Our present study used large numbers of samples, averaging about 35 per species. Compared to laborious measuring of selected lengths as done in traditional morphometrics, data acquisition is far more efficient with a landmark digitization software such as TPSDIG2, which simultaneously captures shape and size variation information. This improved efficiency is important—by greatly reducing the tedium associated with measuring many lengths per specimen, there is more incentive to sample more extensively.

Although the geometric morphometric approach has been strongly advocated by *Vignon & Sasal (2010)* as an effective means to pursue systematics research in monogeneans, the scientific community still lack integrative tools that would make it easy to share data and adopt a common analysis pipeline to ease comparison of old and new results. Here, the monogeneaGM R package that we have developed enables substantial number of shape and size variables from large number of samples to be analyzed efficiently to answer multiple questions, ranging from systematic value of the sclerotized haptoral structures to understanding patterns of phenotypic and phylogenetic correlation. We hope the development of monogeneaGM will contribute to reducing the bottleneck for large scale data analysis in the study of monogeneans. Indeed, as there has been a surge in alpha taxonomy and systematic biology studies of *Ligophorus* in recent years (*Abdallah, De Azevedo & Luque, 2009*; *Siquier & Ostrowski de Núñez, 2009*; *Blasco-Costa, Míguez-Lozano & Balbuena, 2012*; *Dmitrieva et al., 2012*; *Dmitrieva, Gerasev & Gibson, 2013*; *Soo & Lim, 2012*; *Soo & Lim, 2015*; *El Hafidi et al., 2013*; *Kritsky, Khamees & Ali, 2013*; *Sarabeev & Desdevises, 2014*; *Marchiori et al., 2015*; *Rodríguez-González et al., 2015b*; *Soo, Tan & Lim, 2015*), the analysis can only get more interesting as data from other species from other hosts in other geographical regions are added. We expect the data analysis tools in monogeneaGM to continue to evolve to handle complexities of data analysis when this happens.
The use of two-dimensional landmark data implies that the analysis of anchor size and shape evolution is necessarily approximate, since some of the potential biological variation in anchor morphometry may only be adequately captured in three dimensions (*Galli et al., 2007*). Nevertheless, given the wealth of corroborative inference regarding anchor shape and size evolution that have been obtained in the current study, it appears that no general loss of interpretability arises from usage of two-dimensional data for geometric morphometric analysis in *Ligophorus*.

### Future prospects

Adequate taxa coverage remains an important factor for accurate phylogenetic inference (*Sanderson, McMahon & Steel, 2010*). The three markers used in this study are the most common ones reported for other known *Ligophorus* species in the GenBank database, hence their continued use supports efforts at expanding taxa sampling of molecular sequences. Indeed, a research program in *Ligophorus* systematics that expands taxa coverage of anchor geometric morphometric and sequence data opens up the possibility of using parasite phylogeny and anchor morphometry to test hypotheses of host genealogy and ecology (*Nieberding & Olivieri, 2007*) in grey mullets (Teleostei: Mugilidae), a speciose fish family that is economically important (*Durand et al., 2012*). Moreover, analysis of patterns of congruence between the phylogenies of the fish host species and their *Ligophorus* parasites can provide insights into prevalence of host switching (*Zietara & Lumme, 2002*; *Vanhove & Huyse, 2015*) and thence the relative importance of allopatric and sympatric speciation (*Huyse, Audenaert & Volckaert, 2003*) in shaping the diversity of this genus. It is also possible to expand this analysis in a biogeographic context by sampling different geographical populations of a host species, since some *Ligophorus* species have been reported to be useful biological markers of geographical fish host populations (*El Hafidi et al., 2013*). Lastly, we suggest that the present methods of generating and treating geometric morphometric data from the anchors, and possibly other sclerotized hard parts, can be extended to other monogenean families with little difficulty.

## CONCLUSIONS

The presence of significant phylogenetic signal in the anchors makes the quantitative analysis of their shape and size variables useful in answering species discrimination and evolutionary problems in *Ligophorus* specifically, and in other monogenean genera more generally. In this study, we inferred two major host-specific clades from DNA sequence data, which corroborated well with clades inferred from geometric morphometric data of anchors. We further extracted size information through the first principal component of size variables based on all pairwise Euclidean distances between landmarks, and showed that the *Ligophorus* species infecting *Moolgarda buchanani* generally evolved larger anchors compared to those infecting *Liza subviridis*. Anchor shape was correlated with anchor size after controlling for the effect of phylogeny. Subsequently, through the analysis of directional change, we discovered two new morphological characters based on the length between the inner root length and the outer root length, and the length between the inner root point and the dent point, which proved more phylogenetically informative

than existing characters based on the inner length and the outer length. The Rohde-Hobbs hypothesis was validated in the 13 species of *Ligophorus* considered, suggesting the exploitation of different microhabitats and subsequent evolution of reproductive barriers in the form of highly variable male copulatory organ size and morphology after intrahost speciation. Finally, we demonstrated evidence for significant interspecific morphological integration of the root and point compartments within the anchors, as well as integration of the same compartment between the ventral and dorsal anchors.

## SOFTWARE AVAILABILITY

The monogeneaGM package is available for download at https://cran.r-project.org/web/packages/monogeneaGM/. Analyses in this study can be replicated using the R scripts deposited in the Dryad Digital Repository at http://dx.doi.org/10.5061/dryad.50sg7.

## ACKNOWLEDGEMENTS

This work is dedicated to the memory of Lee Hong Susan Lim (1952–2014; see *Gibson & Ng, 2014*), who conceived the present project. Susan Lim contributed actively to global monogenean systematics for decades, and was instrumental at developing and transmitting this art in Malaysia. Klaus Rohde read the early version of the manuscript and provided helpful feedback. We thank Thian Liang Cheow for contributing R codes for data processing. Maarten Vanhove helpfully pointed out some very recent studies that were useful for the Discussion section. Finally, we are grateful to Jean-Lou Justine (Academic Editor), Timothée Poisot, and an anonymous reviewer for their constructive comments which stimulated important improvements to the present work.

### Funding

This work was supported by University of Malaya Research Grant RG197-12SUS to TFK and LHSL, and RP004D-13SUS to LHSL, WBT and TFK. The funders had no role in study design, data collection and analysis, decision to publish, or preparation of the manuscript.

### Grant Disclosures

The following grant information was disclosed by the authors:
University of Malaya Research Grant: RG197-12SUS, RP004D-13SUS.

### Competing Interests

The authors declare there are no competing interests.

### Author Contributions

- Tsung Fei Khang conceived and designed the experiments, performed the experiments, analyzed the data, contributed reagents/materials/analysis tools, wrote the paper, prepared figures and/or tables, reviewed drafts of the paper.

- Oi Yoon Michelle Soo and Wooi Boon Tan performed the experiments, analyzed the data, contributed reagents/materials/analysis tools, wrote the paper, prepared figures and/or tables, reviewed drafts of the paper.
- Lee Hong Susan Lim conceived and designed the experiments.

## DNA Deposition

The following information was supplied regarding the deposition of DNA sequences:

GenBank

KM221934
KM221935
KM221936
KM221937
KM262663
KM221944
KM221945
KM221938
KM221941
KM221942
KM221939
KM221940
KM221943

## Data Availability

http://dx.doi.org/10.5061/dryad.50sg7 Data Dryad Repository link.

## Supplemental Information

Supplemental information for this article can be found online at http://dx.doi.org/10.7717/peerj.1668#supplemental-information.

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

## FURTHER READING

Kahle D, Wickham H. 2013. ggmap: spatial visualization with ggplot2. *The R Journal* **5(1)**:144–161.