# Peer review of "Monogenean anchor morphometry: systematic value, phylogenetic signal, and evolution"

_PeerJ, doi:10.7717/peerj.1668_

## Round 0.1 · original submission · Major Revisions

First, a comment about conflict of interest, since this text will be visible on the PeerJ website. I did had several papers co-authored with one of the authors of this manuscript (Susan Lim), but since she is now deceased, I consider than there are no more possible conflict of interest.

This is an impressive manuscript with a huge number of supplementary files.

I am not a specialist of morphometry but the two reviewers are competent in this field. The paper will probably be accepted when responses have been given to their comments and queries. I will ask the reviewers to review the revised version.

Reviewer 1 ·

Basic reporting

Some sentences in the beginning are worded/constructed in a way that is biologically not very meaningful or clear, like:
- L. 27-28: are anchors only important for monogeneans infecting the gills, or do the authors mean that all monogeneans only infect fish gills? Both seem untrue to me.
- L. 31-32: are hypotheses something that one “answers” to?
- L.39-40, L. 49: I don’t think the authors really delimited species here, nor did they want to do that as these species are all already described. The point was to check whether these species could be distinguished by this analysis.
- L. 139: ecology of the fish host: not only to the ecology, certainly also to its morphology, e.g. the size of the fish’ gill filaments and lamellae plays an important role.

L. 47-48: computational analyses of what?

L. 50-52: do you know of any other study on monogeneans that uses these characters as systematically informative haptoral features, or can you compare this with haptoral characters used for systematic purposes in other studies?

L. 79: please give some examples of monogenean genera with well-resolved phylogenies. Because saying that they are “many” gives me a bit the – unjustified, in my opinion – impression that most (large) monogenean genera are phylogenetically sorted out, which is far from the case, definitely when looking outside of some intensively studied areas like Central Europe.

L. 80-87: when stating that monogeneans are often used as a model, it’d be good to also mention some really recent papers I think, in addition.

L. 127: what do you mean with anchors being the primary opisthaptor attachment organ? The biggest one, the most important one…? Or would you like to offer some functional arguments here?

L. 478: the fact that the tree contains two host-associated clades is in my opinion not necessarily “in agreement with the observed host-specificity”. Species can very well be host-specific in the present, while having undergone host-switches over an evolutionary timescale.

L. 563: I believe the authors’ appreciation of the use of inner and outer root length should be repeated or highlighted in the discussion – it is a very concrete observation to me, of practical relevance to many monogenean workers, and hence should be a bit more in the limelight.

L. 567: such cut-offs are probably only valid within a certain set of species/specimens? How practically usable is the definition of such cut-offs then when trying to distinguish species of any other given monogenean genus?

L. 697-698: I do not think one can make such a comment without explaining to what extent hosts in this region have been sampled for these parasites, to provide the reader with an idea as to how well host range can be estimated at this point.

L. 700-701: I’d like to see some more references that concern monogeneans/flatworms/parasites. I appreciate the use of generally valid texts, but please also don’t forget, in addition, to inform the reader to what extent these concepts have already been explored in your own field of research.

L. 750 and beyond: also a very interesting suggestion! What is this circumstantial evidence exactly, and can the authors ensure that water currents also influence a monogenean in the gill chamber? Just to play the devil’s advocate: isn’t the action of the water sufficiently broken by the mouth/head/gill parts of the fish, doesn’t the operculum provide a certain protection…?

L. 776 and beyond: any idea whether these fenestrations can also be related to muscle attachment?

L. 802 and beyond: the Neotropics are a bit overlooked in this overview, e.g. recent studies from Rodríguez-González et al. (2015, Acta Parasitologica) (Ligophorus from Mexico) and Marchiori et al. (2015, Folia Parasitologica) (Ligophorus from Brazil).

L. 817: indeed genomics offers impressive perspectives, but let’s not forget that these techniques are still very much in their infancy as regards their applicability to monogenean research. There is plenty of literature on the use of molecular markers for (monogenean) flatworms; as it stands now, this statement is a bit of non-sequitur I’m afraid.

Why is there no mention of the relationship between haptoral and genital structures in the Conclusions? Testing the Rohde-Hobbs hypothesis seems quite important, no?

Experimental design

The variety of statistical techniques used in this paper is commendable and impressive.

L. 151: why these particular 13 species, and why mullet hosts? How were target host and parasite taxa decided on?

L. 156 and beyond: “Samples”: how were the fishes collected, and how were they sacrificed?

L.159-160: I suppose “n” stands for the number of hosts caught. You should then probably also mention how many of them are infected and at which intensity.

L. 190-191: since some of the landmarks indicated in Fig. 1 are actually semi-landmarks, I’d add that explicitly to avoid confusion.

L. 206-209: interesting! Without specially treating the semi-landmarks, how did the authors then make sure that the different semi-landmarks represent the same point throughout different individuals? Could the authors elaborate a bit on the controversy regarding the fact that semi-landmarks, when treated in a special way, will distort the result/interpretation? Now it is unclear to me how they, in this analysis, differ from “normal” landmarks, other than the fact that their position is derived from the position of the latter.

L. 221 and beyond: I highly appreciate the use of this quality-checking procedure, which is quite original within the monogenean “world”. It seems to withhold 443 out of 537 slides, hence rejects ca. 20% of the specimens, which seems reasonable to me. However, two remarks/questions with this: (1) How does this incorporate “real” asymmetry between the left and right side structures – asymmetry doesn’t have to be a sign of poor-quality mounts, but can also be a consequence of e.g. environmental stress, isn’t it? (2) Fig. 2 doesn’t really show a poor and a good quality specimen, but their wireframe plots. How can we ascertain whether this doesn’t reflect the definition of the (semi-)landmarks? I think it would be informative to the reader to also show, in addition, a micrograph of the corresponding haptoral hardparts, to show what a specimen of poor/good quality really looks like.

L. 246 and beyond: “Converting Pairwise Euclidean Distances in Arbitrary Units to Physical Units”: this is not clear to me. Perhaps this is a naïve question, but why not directly measure and use the distance in µm? Most image analysis software packages can do this, isn’t it?

L. 282-283: “In doing so, we assumed that the species phylogeny is well-approximated by the phylogeny of these genes…” Ok, we always have to approximate the species tree and hence need such assumptions. However: (1) these markers are of course indeed useful in genetic work on monogeneans, but please cite some of the extensive recent literature discussing the importance and usefulness of the markers (in phylogeny reconstruction, barcoding… what have you) when working with monogeneans, in addition to the very generic reference now provided on the relationship between gene and species trees; (2) in using three adjacent nuclear fragments, the authors didn’t do any particular effort to move away from a gene tree, towards a species tree (e.g. by using markers that involve independently, such as mtDNA fragments). I know of course the complications and the difficulties of amplifying other markers for monogeneans, and I believe in the validity of nuclear rDNA-based results, but please at least mention this limitation and refer to the literature to support your use of these markers.

L. 303-304: it seems important to also mention the length of the fragments that were taken from previous studies.

L. 374-375: the authors refer to a generic reference again to warrant that they check for collinearity. I think it is necessary to also give some more specific information related to the animals under study. E.g., to the best of my knowledge, teasing out the effect of body size on morphological variation is only necessary in animals which continue to grow throughout their lifespan, whence the risk of allometry needs to be avoided, isn’t it? Is this the case in monogeneans? And isn’t there another way to use a kind of standardized proxy for body size, other than the median per species? E.g., in some monogenean taxa there are haptoral structures (i.e. easy to measure) that do not change in adult worms. Could the authors explain why they chose the size of the entire animal, which is hard to compare across mounts, instead to allow a correction for size?

L. 376: I believe the authors mean “body length and body width” instead of “body size and body width”

L. 668: I appreciate mentioning other sources of variation in quality scores. However, could the authors also touch upon whether including a sliding landmark approach, or increasing the number of (semi-)landmarks could have an impact here? In my opinion, the chosen approach for quality control cannot be the only potential solution for the potential confounding factors mentioned.

Validity of the findings

The methods seem sound to me, but I have to admit I am not familiar with several of the statistical approaches used. I recommend to have the paper also reviewed by at least one specialist in statistical data analysis.

L. 177: I would strongly advise that the slides, that seem at the authors’ disposal anyways, be deposited in a curated and permanent institutional collection (e.g. from a natural history museum) to be accessible for future workers.

L. 694: please be aware that the approach of the FishBase Consortium is that, except in the case of meta-analyses, not FishBase should be cited, but rather the primary literature which FishBase uses as a source of the information provided, and which is always clearly cited on FishBase pages.

Additional comments

An interesting paper which includes a variety of approaches and analyses. The piece is nicely written and I enjoyed reading it. Could it be that I detected some minor problems with English, especially in the use of articles?

I appreciate the fact that the authors take the time to dwell on a lot of potential explanations for their findings; however, I do have some critical approaches on some of the conclusions, for which I’d like the authors to provide some more information in the paper – if only also for the benefit of readers less familiar with the approaches used. Please do not see the lengthy comments I make as a negative sign – it is just an extensive paper, with a lot of different aspects, and I would hence like to help the authors to clearly present the substantial amount of information offered.

I believe this paper does a good job and may become an important reference, confirming or refuting, on a formal basis, practices in morphology-based systematics of monogeneans. I would however appreciate if the authors dwell a bit more on the validity of these findings for other taxa of monogeneans – as it appears now, the approach seems mainly valid for dactylogyrideans. I really appreciate the thorough approach of Ligophorus and its hosts, but would like to see the addition of some more excursions towards the potential meaning for other monogeneans (like in the Cichlidogyrus case which was mentioned by the authors). The authors at some points give the impression (e.g. L. 839-840) that they question themselves the usability of this approach to monogeneans other than Ligophorus, and this doesn’t seem necessary to me.

It’d also be good if the authors ensure that the impression is not given that their approach is a “detour” for things which can be handled in a simpler way. E.g., sceptical readers may argue that deciding on how anchors shape and size correlates to host species can also very well be established using simple qualitative or morphometric observations.

Also, I’m a bit surprised that the manuscript does not contain more information on the monogeneaGM package. (What can it do for the reader exactly?) Why not, isn’t this one of the main outcomes/deliverables of the authors’ efforts?

Congrats to the authors for an original study; I hope my comments can be useful to them to clarify some aspects.

·

Basic reporting

The paper is well written, the figures are clear.

I think it would be better to deposit the scripts and data more formally, using e.g. zenodo, rather than having them as supp. mat..

Experimental design

The project is well designed, and the effort of showing how the package intergates with the analysis is appreciable.

Validity of the findings

I wondered how the present findings relate to what we described in Lamellodiscus a few years ago:

http://journals.plos.org/plosone/article?id=10.1371/journal.pone.0026252

Notably, we found that (i) individuals that *looked* the same did not necessarily regroup on the phylogeny and (ii) large-scale groups of morphologies had a weaker than expected phylogenetic signal.

To be clear, I do not think this invalidates the present findings in any way, but since there have been a large number of papers in Lamellodiscus (and Gyrodactylus / Dactylogyrus too), the differences in results deserve to be discussed. Is there something in the parasite ecology that explains this difference? Are Ligophorus more specialized than some of the Lamellodiscus?

---

## Round 0.2 · Minor Revisions

My apologies for the delay: one of the reviewers was travelling and asked for additional time.

Both reviewers - and the Editor! - were impressed by the quality of the answers to the comments.

One reviewer suggested to add some of your answers in the manuscript itself. I believe it is a useful suggestion. Please edit the manuscript accordingly and send it back - I will be happy to accept it without further delay.

Reviewer 1 ·

Basic reporting

no comments

Experimental design

no comments

Validity of the findings

no comments

Additional comments

Dear authors, dear editor

Thank you for the opportunity to review this revised MS. I have gone through MS with particular emphasis to the changes made, and find them very convincing. I have no doubts regarding the authors' statements and the issues where
I had questions are now all clarified satisfactorily.

Some minor suggestions only:
1) I'd like to see (at least part of) the replies to R1.20 and R1.21 included in the study, as these sentences contain a lot of methodological information that highlight the importance and clarify the approach of this study, making it more understandable
to non-specialists, such as:
- the proportion of species within Ligophorus that are covered by this study (ca. 20% is not bad at all!);
- the importance of these fishes as food source (e.g. in my own region, they do exist but are hardly eaten and hence less well-known with non-ichthyologists or non-Ligophorus connoisseurs);
- the fact that your coverage of Ligophorus is apparently exhaustive with regards to the species known from these hosts (very good!);
- the fact that fish were collected from markets is important to let the reader form an idea of how you carried out one of the first, and very important, practical steps of such a study: sampling.

2) R1.32: please mention the availability/deposition of these specimens in these museum collections somewhere in the main text;
the readers can then always refer to the supplementary material when they want to know the accession numbers etc., but may not know
that they are available anyways when this fact is not mentioned in the main text.

3) L. 128 in revised MS: "defined points" - I do not think a dash is needed here

4) L. 137 in revised MS: should be "synonymies"

Once again, congratulations with this important work, and I am looking forward to the authors' continued contributions to monogenean biology.

·

Basic reporting

N/A

Experimental design

N/A

Validity of the findings

N/A

Additional comments

These are the most impressively thorough replies to reviewers I have ever seen. It took me some time to go through all of them, but after having done so, I have no remaining issues.

---

## Round 0.3 · accepted · Accept

No further comments! The last modifications are good.